# The Life Cycle of Upper-Level Troughs and Ridges: A Novel Detection Method, Climatologies and Lagrangian Characteristics

Sebastian Schemm [1], Stefan Rüdisühli [1], and Michael Sprenger [1]

[1]Institute for Atmospheric and Climate Science, ETH Zurich, Zurich, Switzerland

**Correspondence:** Sebastian Schemm (sebastian.schemm@env.ethz.ch)

**Abstract.** A novel method is introduced to identify and track the life cycle of upper-level troughs and ridges. The aim is to close the existing gap between methods that detect the initiation phase of upper-level Rossby wave development, and methods that detect Rossby wave breaking and decaying waves. The presented method quantifies the horizontal trough and ridge orientation and identifies the corresponding trough and ridge axes. These allow us to study the dynamics of pre- and post-trough/ridge regions separately. The method is based on the curvature of the geopotential height at a given isobaric surface and is computationally efficient. Spatio-temporal tracking allows us to quantify the maturity of troughs and ridges, and could also be used to study the temporal evolution of the trough or ridge orientation. First, the algorithm is introduced in detail, and several illustrative applications – such as a downstream development from the North Atlantic into the Mediterranean – and seasonal climatologies are discussed. For example, the climatological trough and ridge orientations reveal strong zonal and meridional asymmetry: over land, most troughs and ridges are anticyclonically oriented, while they are cyclonically oriented over the main oceanic storm tracks; the cyclonic orientation increases toward the poles, while the anticyclonic orientation increases toward the equator. Trough detection frequencies are climatologically high downstream of the Rocky Mountains and over East Asia and Eastern Europe, but are remarkably low downstream of Greenland. Furthermore, the detection frequencies of troughs are high at the end of the North Pacific storm track, and at the end of the North Atlantic storm track over the British Isles. During El Niño-affected winters, troughs and ridges exhibit an anomalously strong cyclonic tilt over North America and the North Atlantic, in agreement with previous findings based on traditional variance-based diagnostics such as $\mathbf{E}$ vectors. During La Niña the situation is essentially reversed. The orientation of troughs and ridges also depends on the jet position. For example, during midwinter over the Pacific, when the subtropical jet is strongest and located farthest equatorward, cyclonically oriented troughs and ridges dominate the climatology. Finally, the identified troughs and ridges are used as starting points for 24-hour backward parcel trajectories, and a discussion of the distribution of pressure, potential temperature and potential vorticity changes along the trajectories is provided to give insight into the three-dimensional nature of troughs and ridges.

## 1 Introduction

Troughs and ridges are ubiquitous flow features in the upper troposphere and are centerpieces of weather and climate research, for good reason. In general, a trough is associated with cyclonic flow and moving cold air equatorward. It is depicted as a region of reduced geopotential height on an isobaric surface, or enhanced potential vorticity on an isentropic surface. The counterpart

of a trough is a ridge, which is associated with warm air moving poleward, increased geopotential height – corresponding to reduced potential vorticity (PV) – and anticyclonic flow. Jointly, a trough and a ridge form the positive and negative phases, respectively, of large-scale Rossby wave patterns, which shape weather development in the mid-latitudes. At the equatorial tip of a trough, where the geopotential isolines are closely aligned, a jet streak can form, and the conservation of the absolute

vorticity and a region of diffluent flow predict forced upward motion in the jet exit region. It is therefore not too surprising that surface cyclogenesis (Petterssen and Smebye, 1971; Sanders, 1988; Lackmann et al., 1997; Graf et al., 2017), rapid cyclone intensification (Sanders and Gyakum, 1980; Wash et al., 1988; Uccellini, 1990; Wernli et al., 2002; Gray and Dacre, 2006) and enhanced precipitation (Martius et al., 2006; Massacand et al., 2001; Martius et al., 2013) usually take place in the region ahead of the upper-level trough axis.

The shape and orientation of troughs and ridges is pivotal in determining their influence on synoptic-scale flow evolution. A trough or a ridge tilts cyclonically if it forms under cyclonic shear, while it acquires an anticyclonic orientation if it forms under anticyclonic shear (Davies et al., 1991; Thorncroft et al., 1993). In the case of a pronounced equatorward (poleward) excursion of a trough (ridge), it may wrap up and undergo irreversible deformation in a wave breaking event (McIntyre and Palmer, 1983; Thorncroft et al., 1993). Under anticyclonic shear, the trough deforms into a narrow band, called a streamer, which crosses the

jet and extends equatorward of the mean jet position and cut-off formation may occur. During the associated wave breaking event, the jet is pushed poleward. With cyclonic shear, wave breaking occurs poleward of the mean jet position, thereby pushing the jet equatorward (Thorncroft et al., 1993; Lee and Feldstein, 1996; Orlanski, 2003). The notion of cyclonically and anticyclonically oriented wave development is schematically summarized in Fig. 12 in Thorncroft et al. (1993). This process, through which cyclonically or anticyclonically breaking troughs and ridges displace the jet, is even able to excite

positive or negative North Atlantic Oscillation events (Benedict et al., 2004; Franzke et al., 2004; Rivière and Orlanski, 2007). A traditional means for quantifying the influence of upper-level troughs and ridges on the jet strength and location is the **E** vector (Hoskins et al., 1983; Trenberth, 1986). With anticyclonic shear, troughs and ridges acquire an anticyclonic orientation, and the **E** vector points equatorward, indicating poleward eddy momentum flux. With cyclonic shear, a poleward-pointing **E** vector indicates southeast-to-northwest-oriented troughs and ridges, which correspond to a cyclonic orientation, and the

corresponding equatorward eddy momentum flux pushes the jet equatorward (Hoskins et al., 1983; Rivière et al., 2003).

Troughs and ridges actively interact with diabatic processes and are far from being dry adiabatic flow features. Ahead of the trough axis, where upward vertical motion prevails, air from the surface warm sector of a cyclone is lifted within the warm conveyor belt (WCB) to upper levels (Harrold, 1973; Browning, 1986, 1990; Wernli, 1997). During ascent, the air is diabatically modified – for example by condensation – in a way that amplifies the upper-level ridge and the anticyclonic flow downstream

of the surface cyclone (Stoelinga, 1996; Wernli and Davies, 1997; Grams et al., 2011; Schemm et al., 2013). This process also amplifies the streamer formation (Massacand et al., 2001) and accelerates the jet. From a PV perspective, this phenomenon is seen as a negative PV anomaly around the WCB outflow, corresponding to enhanced anticyclonic circulation. It is also associated with a sharpening of the PV gradient along the edge of the WCB outflow, which corresponds to an acceleration of the jet. From a momentum flux perspective, the center of the WCB outflow corresponds to a region of enhanced horizontal eddy

momentum flux divergence (leading to a deceleration), and the edge of the WCB outflow corresponds to a region of enhanced

eddy momentum flux convergence (leading to an acceleration) (Schemm, 2013). The WCB outflow ahead of an upper-level trough has thus been shown to accelerate downstream cyclone growth (Wernli and Davies, 1997; Pomroy and Thorpe, 2000; Grams et al., 2011; Schemm et al., 2013) and can even lead to the formation of large-scale blocks (Pfahl et al., 2015; Steinfeld and Pfahl, 2019). Recent studies have presented further evidence for the non-negligible role of turbulence and radiation in the diabatic modification of the life cycle of troughs and ridges (Spreitzer et al., 2019). The dynamics of troughs and ridges thus may act as a stepping stone toward a better understanding of the coupling between adiabatic and diabatic processes and allows for the connection of atmospheric processes from the planetary scale to the mesoscale.

In this study, a novel feature-based method is introduced for identifying and tracking the life cycle of upper-level troughs and ridges, including their axes and orientations, in gridded data. Feature-based methods for detecting streamers (Wernli and Sprenger, 2007), Rossby wave initiation (Röthlisberger et al., 2016), and wave breaking (e.g., Postel and Hitchman, 1999) events are widely used research tools, and the motivation behind this work is to extend the capability of available tools and to track and characterize the entire life cycle of upper-level troughs and ridges from genesis to lysis. We further aim to compare the detected life cycle characteristics of, for example, the trough orientation with previous results obtained from more traditional methods such as the aforementioned **E** vectors, thereby bridging gaps between different perspectives on synoptic-scale evolution.

The trough and ridge detection is used to derive winter and summer climatologies of their frequency and orientation. Further, we address two research questions discussed in the recent literature. The first concerns the change in the orientation of downstream propagating eddies during winters affected by the El Niño–Southern Oscillation (ENSO): During La Niña, eddies are assumed to acquire a more anticyclonic orientation over North America, while they acquire a more cyclonic orientation during El Niño (Li and Lau, 2012a, b; Drouard et al., 2015). Previous studies used the **E** vector (Hoskins et al., 1983; Trenberth, 1986) to study the eddy orientation. Here, we quantify the trough and ridge tilt using the automated algorithm. The second research question concerns the midwinter suppression of the North Pacific storm track: During midwinter, the potential for cyclone growth is largest, but – in contrast to the North Atlantic – the observed storm track intensity over the Pacific is lower compared to the shoulder seasons (e.g., Nakamura, 1992). Using the trough and ridge detection, we explore potential changes in the orientation of troughs and ridges, because a stronger cyclonic orientation is typically associated with more intense cyclones (e.g., Fig. 4 in Thorncroft et al., 1993). More detailed literature overviews are presented in the corresponding sections. Finally, we explore the possibility to use the trough and ridge objects as starting regions for air parcel trajectories in an effort to explore the diabatic modification of the life cycle of troughs and ridges.

The paper is organized as follows. Details of the algorithm and a case study are presented in Section 2. Section 3 is dedicated to the discussion of winter and summer climatologies and the change in the trough and ridge orientation during ENSO-affected winters and over the Pacific in midwinter. In Section 4, we explore troughs and ridges from a Lagrangian viewpoint before concluding the paper in Section 5.

## 2 Data and Methods

The trough and ridge identification algorithm uses 6-hourly ERA-Interim data (Dee et al., 2011) but can easily be applied to other gridded data sets. The reanalysis data are provided publicly by the European Centre for Mid-Range Weather Forecasts (ECMWF) via URL <apps.ecmwf.int/datasets>. More specifically, the identification is demonstrated on the 500 hPa geopotential height data, which are interpolated onto a regular $1° \times 1°$ grid. Results based on the 300 hPa geopotential height data are shown in a supplement. The 500 hPa geopotential height has traditionally been used for analysis of troughs and ridges and continues to be used for research and forecasting purposes. The presented method is, in general, also applicable to different levels or variables, e.g., 300 hPa geopotential height, isentropic PV or potential temperature on the dynamical tropopause. Depending on the exact research question the most suitable level and variable can be chosen. The investigation period is 1979–2018, and the monthly mean trough and ridge climatologies are publicly available [eraiclim.ethz.ch; Sprenger et al. (2017)].

This section describes in detail how the trough and ridge axes are identified and tracked. Three distinct steps are involved:

– identifying 2D trough and ridge objects (masks) and their corresponding axes;

– tracking the trough and ridge objects and quantifying their age and overall lifetimes;

– characterizing the horizontal orientation of each trough and ridge axis.

The following subsections will address specific aspects of the algorithm, including a detailed validation.

### 2.1 Trough and Ridge Identification

The identification of troughs and ridges starts with the geopotential height at, for example, 500 hPa, as seen in Fig. 1a for 12 UTC on 12 January 2010. Two distinct troughs and one ridge can visually be identified: a northwest-to-southeast oriented trough over the eastern North Atlantic, which is associated with a mature low pressure system to the west of Ireland; a meridionally oriented ridge, which extends from the Iberian Peninsula to the British Isles; and a meridionally oriented trough over the Central Mediterranean to the southeast of Italy over the Ionian Sea. The goal of an object-based trough and ridge detection is to automatically identify these features and the corresponding axes. Several aspects have to be kept in mind. First, the smooth geopotential isolines shown in Fig. 1a do not reflect the underlying $1° \times 1°$ ERA-Interim grid, but are already smoothed due to the contour drawing algorithm. Therefore, in the following panels (Fig. 1b to f), the geopotential height is shown on the underlying input grid. Second, troughs and ridges may exhibit a complicated structure. For instance, the horizontal orientation can vary along a trough or ridge axis; a trough or ridge can be broken into different segments; and the curvature of the geopotential isolines can considerably vary over small distances. The challenge for an automatic detection algorithm is to identify all these different characteristics. In the following, details of the proposed algorithm are described.

We suggest identifying troughs and ridges geometrically using the curvature of the geopotential isolines. To this end, the change of the orientation of a vector pointing along an isoline is computed. First, at every grid point, the gradient of the geopotential field is calculated and the gradient vector is rotated by $90°$, such that the rotated vector aligns with the geopotential isolines. Next, a new location 5 km in the direction of the aligned vector is determined, and the corresponding aligned vector

at this new location is obtained trough bilinear interpolation. Because of the curvature of the geopotential isolines, the original vectors and those shifted by 5 km are rotated against each other, and it is the angle between these two vectors – normalized by the distance of 5 km – that we define as the local curvature. The curvature field $\alpha(x, y)$ is available on the whole global grid and is used to identify the trough and ridge objects. These steps are summarized in Fig. 2.

5      To identify troughs and ridges, a threshold of $5 \cdot 10^{-5}$ degrees per meter is used to mask the curvature field (Fig. 1b). This threshold is the main degree of freedom of the algorithm. All connected points in the masked field, which form a coherent object, are clustered into one distinct 2D trough or ridge depending on the sign of the curvature field (Fig. 1c). Grid points outside a coherent object are flagged with zeros, grid points insides with ones. Time-averaging over the obtained binary fields yields detection frequencies, which indicate the fraction of time steps affected by a trough or ridge relative to all time steps. We 10   selectively disregard very small features ($< 20 \cdot [111\,\text{km}]^2$), which at this stage are unlikely to correspond to a significant flow deviation. Additionally, consideration is only given to midlatitude troughs and ridges within 20°N–70°N, but the algorithm in general could be applied also to polar latitudes. Figure 1 illustrates the individual steps in more details. Figure 1b shows the trough and ridge objects after masking the normalized curvature field using the above threshold. The trough over the eastern North Atlantic is connected to a mature low-pressure system, which is discernible from the closed 500 hPa isolines in the 15   upper-left corner of Fig. 1a and which is also well-marked at the surface. This trough is classified as a "closed trough". On the other hand, the trough object over the Central Mediterranean is not associated with a low-pressure system and is therefore classified by the algorithm as an "open trough". We decided to classify regions inside closed geopotential contours not as part of a trough or ridge object – as is the case for the trough over the Atlantic in Fig. 1b – but rather flag the region inside a closed geopotential contour as a distinct low- or high-pressure system, as in Sprenger et al. (2017). In this way, troughs and ridges 20   can be classified as either independent of, or as linked with, a low- or high-pressure system; and the troughs and ridges are classified as such by the algorithm.

     In a final step, the corresponding trough and ridge axes are identified. The result of this step is shown in Fig. 1d. Algorithmically, a starting point is first selected within each 2D trough and ridge object; we decided to start at the highest (for troughs) and lowest (for ridges) geopotential value within each object. From this starting point, a line is iteratively constructed by stepping 25   5 km forward parallel to the geopotential gradient. The iterative extension of the axis line ends as soon as it leaves the 2D trough or ridge object. If the identified axis is shorter than 500 km, the axis and the corresponding 2D trough or ridge object are removed. This minimum length is the second degree of freedom in our algorithm that the user can adjust. The final outcome of the trough and ridge identification is therefore a gridded field, corresponding to the original input grid, on which trough and ridge grid cells are labeled and flagged in different ways (e.g., open or closed). For plotting purposes, it is convenient to 30   transform the axes into 1D polylines by means of a cubic-spline interpolation (as is done in Fig. 1e,f).

     The troughs and ridges are further characterized with respect to their horizontal orientation. The horizontal orientation (or tilt) of a trough or ridge differs from the curvature of the geopotential isolines. To obtain it, the angle of the trough or ridge axis is estimated relative to a north-south meridian at every grid point and projected laterally to the whole 2D object. We refrain from attributing a unique orientation to the entire 2D trough or ridge object. Instead, the orientation can change, or even reverse its 35   sign, along the axis of a trough or ridge. An example is shown in Fig. 1f. A positive angle corresponds to a southwest-northeast

orientation (anticyclonic), a negative angle to a southeast-northwest orientation (cyclonic), and the orientation can change within the same object. Near-zero values correspond to meridionally oriented troughs and ridges. For instance, the southeast-northwest oriented trough in the North Atlantic is tilted cyclonically by 30–45° relative to the north-south meridians (Fig. 1f). On the other hand, the ridge to the south of the British Isles is very weakly tilted relative to the meridians, and marginally changes its orientation from its northern to its southern tip. The easternmost trough over the Central Mediterranean has a mild anticyclonic orientation (blue shading in Fig. 1f).

Finally, we apply a tracking algorithm to determine the age and overall lifetime of the individual trough and ridge objects. An example is shown in Fig. 1e, where the trough over the eastern North Atlantic is found to be rather young (6–12 h) – a little older than the trough over the Central Mediterranean (0–6 h), but younger than the ridge to the south of the British Isles (18–24 h). Algorithmically, the tracking determines temporal connections between trough or ridge objects while naturally accounting for mergings and splittings. The connections between two consecutive time steps, $(t_i)$ and $(t_j)$, are found as follows. First, each object at either time step is paired with each unique combination of up to three overlapping objects at the other time step. For example, if object $A(t_i)$ at time step $t_i$ overlaps objects $B(t_j)$ and $C(t_j)$ at $t_j$, three connections are possible: $A(t_i) \rightarrow B(t_j)$, $A(t_i) \rightarrow C(t_j)$, and $A(t_i) \rightarrow B(t_j) + C(t_j)$, whereby the latter would constitute a splitting if $t_j > t_i$ and a merging if $t_i > t_j$. Second, each of these potential connection is assigned a probability, which is high if the combined object size at the two time steps is similar and the involved objects exhibit substantial overlap over time. Third, the final connections are selected iteratively in descending order of probability, whereby each object can only be part of one connection. Objects for which no connection is found constitute the beginning or end of a track (or branch thereof). Further details can be found in Section 2.2 of Rüdisühli (2018). We use the tracking to identify young troughs or ridges, but it could also be used, for example, to study the change in the orientation of the troughs and ridges during their life cycle.

## 2.2 Illustrative case: Downstream development over the North Atlantic–Mediterranean sector

In this section, an illustrative example is briefly discussed. Figure 3 compares two distinct time instances (06 UTC and 12 UTC on 12 January 2010, in Fig. 3a and b, respectively) for a synoptic evolution over the eastern North Atlantic–Mediterranean sector. At 06 UTC on 12 January 2010 (Fig. 3a), a trough – which is connected to a mature low-pressure system over the eastern North Atlantic – is located near 20°W. Quasi-geostrophic (QG) forcing for downward motion (yellow contours in Fig. 3) is found upstream of the trough axis, and forcing for upward motion (green contours in Fig. 3) on its downstream side. The forced omega is computed at 500 hPa according to the Q-vector formulation of the classical quasi-geostrophic omega equation in pressure coordinates,

$$\sigma \nabla^2 \omega + f_0^2 \frac{\partial^2 \omega}{\partial p^2} = -2 \nabla \cdot \mathbf{Q}, \tag{1}$$

where $\sigma$ denote the static stability, $f_0$ the Coriolis paramter, $\omega$ the vertical motion and $\mathbf{Q}$ the Q-vector. For details of the derivation we refer to Holton (2004), and for details of this computation we refer to the supplement of Graf et al. (2017). The identified trough axis sits in the transition zone between downward and upward forcing, highlighting the physical relevance of the identified axis, and therefore could be used to separate the pre-trough from the post-trough sector. A downward and upward

QG-forcing pattern is also discernible over the Central Mediterranean, but a corresponding trough axis is missing at 06 UTC (Fig. 3a), which is related to the degree of freedom the user has in setting the minimum curvature and length of the axis. Six hours later, at 12 UTC (Fig. 3b), the trough axis is identified and located in-between the upward and downward forcing. Overall, the temporal evolution of the synoptic situation is reminiscent of the situations discussed by Raveh-Rubin and Flaounas (2017).

It seems to follow a common pattern for Mediterranean cyclogenesis: The outflows of warm conveyor belts from upstream-cyclones over the Atlantic Ocean tend to amplify a ridge over the eastern Atlantic, and the consequent downstream wave development intrudes the Mediterranean in southern latitudes, provoking cyclogenesis. Interestingly, the ridge downstream already exists for a longer time period than the up- and downstream troughs. This tends to be in agreement with the finding in Raveh-Rubin and Flaounas (2017) that a series of Atlantic cyclones is necessary to initiate Mediterranean cyclogenesis.

The trough and ridge tracking could be used to quantify the time between the formation of the upstream Atlantic trough, the downstream ridge, and the downstream Mediterranean trough, plus the typical orientation of these features preceding Mediterranean cyclogenesis.

## 3   Climatologies

In this section, the trough and ridge diagnostics presented above are applied to compute climatologies for the extended winter

(Nov–Mar) and summer (May–Sep) seasons 1979–2018 in the Northern Hemisphere. We restrict the discussion for practical reasons to the extended winter and summer seasons and the 500 hPa level, as we intend to present an in-depth discussion of the monthly and seasonal cycles in both hemispheres in future publications. The results for the 300 hPa level are shown in the supplement.

### 3.1   Extended Winter (November–March)

In Fig. 4a, a summary is given for the frequency of trough objects, the frequency of incipient trough age (yellow contour in Fig. 4a), and the mean horizontal orientation (Fig. 4b) at the 500 hPa level. During winter, the trough detection frequency displays four centers of action around the Northern Hemisphere, for example over North America downstream of the Rocky Mountains – a well known surface cyclogenesis region (e.g., Bannon, 1992; Hobbs et al., 1996; Hoskins and Hodges, 2002) – and downstream of the Altai. The trough detection frequency also peaks at the end of the North Pacific storm track over the Bay

of Alaska, in agreement with climatologies of surface cyclones (Fig. 4a in Wernli and Schwierz, 2006), cyclolysis (Fig. 5d in Hoskins and Hodges, 2002) and PV streamers on several isentropic levels (310–330 K in Fig. 3 in Wernli and Sprenger, 2007). It also peaks over Eastern Europe to the north of the Black Sea, which is seen in streamer climatologies at lower isentropic levels (310 K in Fig. 3b of Wernli and Sprenger, 2007). The latter maximum has an upstream branch into the Mediterranean and a downstream branch into the Caspian Sea. Surprisingly – and in contrast to the Rocky Mountains – there is no peak downstream

of Greenland, which is an important surface cyclogenesis region. Greenland surface cyclogenesis is typically preceded by an eastward-propagating upper-level trough-ridge train, which is seen, for example, in Fig. 4 of Schemm et al. (2018). Because detection frequencies tend to highlight regions where troughs are stationary, it seems as if troughs downstream of Greenland

are more transient compared to their counterparts downstream of the Rocky Mountains. Indeed, the mean lifetime of troughs is lower at Greenland compared to downstream of the Rocky Mountains (Supplementary Fig. S4) Furthermore, there is no center of action over the Nordic Seas, which is one exit region of the North Atlantic storm track. This is in contrast to the exit of the North Pacific storm track and the maximum over the Bay of Alaska.

While all main upper-level storm tracks that are identified in earlier studies (Hoskins and Hodges, 2002) are also successfully identified with our approach, there is surprisingly no strong signal discernible along the southern branch over Eurasia highlighted by Chang and Yu (1999) and Chang (2005). In principal, the algorithm is capable of picking up troughs and ridges near 25°N over Eurasia, as we confirmed by a manual inspection of cases, but the detection is less frequent as one might expect. Potential reasons for this observation are discussed in the caveats section at the end of the article.

The trough frequencies of incipient troughs are obtained using the tracking capability of the algorithm (section 2). To this end, all trough objects that are older than one day are removed from the data, and the detection frequencies of these early troughs are re-computed based on the remaining trough objects. These incipient troughs (yellow contour in Fig. 4a indicates half of the maximum value in early-trough detection frequency) are frequently detected downstream of the Rocky Mountains and downstream of the Altai, suggesting that these two regions are preferred trough genesis regions. There is also a smaller

peak in the early-trough frequency near Greenland, suggesting that trough genesis also occurs near Greenland. However, the reduced trough frequency compared to the Rocky Mountains suggests that Greenland troughs are indeed more transient. A second small peak of early troughs is seen in the Gulf of Genoa, a preferred region for Mediterranean cyclogenesis (Trigo et al., 2002). There are two additional, broader regions where early troughs are frequent and not linked to orography: the end of the storm track over the North Pacific, and over parts of Northern Europe extending from the British Isles eastward across

Central Europe into Russia. It is plausible to think of these early troughs over Europe as a result of synoptic systems that decay upstream over the eastern North Atlantic. Wave breaking and consecutive downstream developments at the end of a synoptic wave life cycle over the eastern North Atlantic provide then the seed for trough genesis further downstream over Europe. Finally, a note of caution regarding the climatological trough age is in order. To obtain the age climatology, the age information obtained from the trough tracking is assigned to every grid point inside a two-dimensional trough object, while points outside

are labeled as missing data. Time averaging of this field results in the mean age at each grid point. The climatological age is therefore a function of the trough size, and the age contours therefore enclose smaller regions where troughs tend to be small. Furthermore, the number of troughs at each grid point varies according to the trough frequency (color shading in Fig. 4a); and near the lateral boundaries of the domain, a very low number of troughs dictates the climatological mean. Late-trough frequencies typically encompass larger regions and extend further downstream (not shown), but the absolute frequency values

are considerably reduced.

The trough orientation displays a strong zonal asymmetry. Over the main oceanic storm tracks, the mean trough orientation is preferentially cyclonic (blue shading in Fig. 4b). The cyclonic orientation increases toward the end of the storm tracks and also with latitude. Over land, the trough orientation is preferentially anticyclonic. The anticyclonic orientation increases toward lower latitudes. This meridional dependence of the mean trough orientation is in agreement with the conventional interpretation of cyclonic and anticyclonic wave life cycles and the associated wave breaking, which occurs poleward (cyclonic)


or equatorward (anticyclonic) of the mean jet position (Thorncroft et al., 1993). Over Northern Africa, the climatological trough orientation is strongly anticyclonic. In this region, troughs are frequently associated with anticyclonic wave breaking downstream of a mature extratopical cyclone situated off the Iberian Peninsula. This downstream trough eventually thins and elongates equatorward, corresponding to PV streamer formation. Eventually, an upper-level cutoff low forms. From the perspective of Rossby wave packets propagating along wave guides, the anticyclonic troughs in this sector have been described as the transmitters between wave packets initiated on the subtropical wave guide (i.e., the jet over Northern Africa) by wave packets that propagate along the extratropical wave guide over the North Atlantic (Martius et al., 2010, in particular their Fig. 5).

In Fig. 5a, a summary is given for the number of ridges detected during the cold season. Remarkably, the ridge detection frequency over many regions is larger by almost a factor of two compared to the trough detection frequency. For example, the ridge detection frequency downstream of Greenland with a preferred cyclonic orientation (Fig. 5b) is twice as high as that for troughs. The band of enhanced ridge frequencies elongates downstream over the Nordic Seas, peaks over the Scandinavian Mountains and remains at a relatively high level over Siberia. Upstream of the Altai, the ridge frequency decreases, while downstream, the trough frequency peaks (Fig. 4a). Over large parts of Siberia, the ridges have no preferred orientation, while they tend to be anticyclonically oriented over East Asia. This band of higher ridge frequencies might relate to the Siberian storm track – seen in the track densities of 250 hPa meridional wind anomalies (Fig. 4a in Hoskins and Hodges, 2019) – while the main surface storm track is located further poleward over the Barents Sea region (Fig. 4a in Wernli and Schwierz, 2006). Over East Asia and the western Pacific, the ridge detection frequency is rather low, while the climatology over the Coast Mountains of western North America is dominated by stationary ridges which preferentially have a cyclonic orientation (blue shading in Fig. 5b). Downstream of the Rocky Mountains ridge frequency is rather low. A smaller region of high early-age ridge frequencies is found near Kamchatka (yellow contour in Fig. 5a), which indicates the presence of a transient ridge with a preferred cyclonic orientation (blue shading in Fig. 5b). Indeed, a local maximum in the surface cyclone frequencies is found downstream of Kamchatka (see Fig. 4a in Wernli and Schwierz, 2006). Overall, the trough and ridge pattern alternates almost periodically in terms of the amplitude around the Northern Hemisphere.

We further explored the role of the level and the minimum lifetime of the detected features on the trough and ridge frequencies. The trough and ridge detection patterns and the corresponding orientation at the 300 hPa level are in very close agreement with the results at the 500 hPa level (Supplementary Fig. S1). The detection frequencies are increased everywhere by 10–20 %, but the main centers of action and the mean orientation agrees well between the levels. We further computed the climatologies after removing short-lived features that have a maximum lifetime of 24 hours in a post-processing step (Supplementary Fig. S3). Again, the resulting patterns parallel those shown in Fig. 4 and Fig. 5, but the detection frequencies are reduced.

### 3.2 Extended Summer (May–September)

The pattern in the trough detection frequencies changes from winter to summer (Fig. 4). Over North America, the main frequency peak is located downstream of the Great Lakes Region and over Newfoundland, while it was previously located further upstream. This is in agreement with the high surface cyclone frequencies seen in summer in this region (see Fig. 4c in

Wernli and Schwierz, 2006). Over the North Atlantic, the main frequency peak is located just off the Iberian Peninsula. There is no comparable peak in surface cyclone frequencies in this region, so the flow conditions will result in the formation of a cut-off low with no well-marked surface signature. Surface cyclone frequencies have a peak downstream of Greenland (see Fig. 4c in Wernli and Schwierz, 2006), but there is no maximum trough frequency there. However, upper-level storm-track

measures – based on, for example, track densities of 250 hPa vorticity – indeed indicate the presence of a trough-like feature off the Iberian Peninsula during summer (see Fig. 1c in Hoskins and Hodges, 2019). Because a similar feature is also seen in summertime climatologies of Rossby wave breaking (see Fig. 10a in Postel and Hitchman, 1999), we expect this maximum in trough frequency to be related to anticyclonic Rossby wave breaking as described for the LC1 scenario in Fig. 12a of Thorncroft et al. (1993). An example of such synoptic situation is also shown in Fig. 5.7 of Martius and Rivière (2016).

Over Eurasia, a maximum in the summertime detection frequencies of troughs connects the Eastern Mediterranean with the Black Sea. This feature is not well reproduced in Rossby wave breaking climatologies of which we are aware, but it is clearly seen in the upper-level track densities in vorticity on 250 hPa (Hoskins and Hodges, 2019). Furthermore, this local maximum corresponds to a similar maximum seen in summertime climatologies of stratospheric PV streamers (see Fig. 6b in Wernli and Sprenger, 2007). PV streamers form a subcategory of troughs and are best described as filament-like elongated troughs,

which have a length that is longer than their width (Wernli and Sprenger, 2007). Streamers are associated with anticyclonic Rossby wave breaking, so it is not too surprising that the detection hot spots off the Iberian Peninsula and near the Black Sea eventually become part of the same Rossby wave train [see for example Fig. 3 in (Wernli and Sprenger, 2007) for such a synoptic situation]. A second maximum over Eurasia is located west of 90°E, a feature seen again in the streamer climatology of Wernli and Sprenger (2007). Further east, the maximum over East Asia exhibits only weak changes in its location between

winter (Fig. 4a) and summer (Fig. 4c) but appears with a reduced amplitude during summer. Finally, over the northeastern Pacific, the summer maximum is shifted slightly equatorward compared to its position during winter. It is located off the US west coast during summer, while it is located in the Bay of Alaska during winter. This is a counter-intuitive result because storm tracks generally tend to shift poleward during summer (Hoskins and Hodges, 2019).

The ridge frequency pattern over North America continues to be dominated by the stationary ridges over the Coastal Moun-
tains despite a mild reduction in the absolute detection frequencies (color shading in Fig. 5c).

Finally, we note that – as is the case for the winter season – the results obtained at the 300 hPa level (Supplementary Fig. S1) are in very close agreement with those at the 500 hPa level. The 300 hPa detection frequencies are moderately higher, but the patterns are remarkably similar.

### 3.3  ENSO-affected winter seasons

The influence of the ENSO on the midlatitudes is a longstanding research topic – for recent reviews, we refer the reader to publications by Liu and Alexander (2007); Stan et al. (2017) and Yeh et al. (2018). Recently, a particular progress has been made in understanding the role of synoptic-scale eddies in shaping the North Atlantic circulation response to ENSO. More specifically, there is increasing evidence that the North Atlantic teleconnection pattern is in part a downstream response of the eddy-driven jet to changes in the orientation of synoptic-scale eddies entering the North Atlantic from North America (Li and

Lau, 2012a, b; Drouard et al., 2015). The mechanisms can be briefly summarized as follows – for a schematic summary see also Fig. 13 in Schemm et al. (2018): In response to an amplified ridge over the northeastern Pacific during La Niña, upper-level synoptic eddies with a more anticyclonic orientation form downstream of the Rocky Mountains, where lee cyclogenesis is also enhanced. These eddies propagate downstream over the North Atlantic while maintaining their anticyclonic orientation until anticyclonic wave breaking occurs over the eastern North Atlantic. Anticyclonic wave breaking pushes the eddy-driven jet poleward. Until now, the anomalous anticyclonic orientation and the associated more poleward eddy momentum flux were diagnosed using the horizontal **E** vectors of Hoskins et al. (1983) and Trenberth (1986), which are obtained using high-pass filtered wind data,

$$\mathbf{E} = \begin{bmatrix} \frac{1}{2}\left( \overline{v^{*2} - u^{*2}} \right) \\ -\overline{u^* v^*} \end{bmatrix}, \tag{2}$$

where the overbar indicates a time average, and the asterisks a deviation from the time average. The fundamental relationships between the trough and ridge orientation and the eddy momentum flux have been discussed as early as by Jeffreys (1926) and Starr (1948). As summarized in the introduction, equatorward-pointing **E** vectors are assumed to indicate anticyclonically oriented eddies (Rivière et al., 2003) associated with a more poleward eddy momentum flux. During El Niño, the situation is essentially the opposite: Eddies downstream of the Rocky Mountains exhibit a more cyclonic orientation and tend to push the North Atlantic jet equatorward, as is suggested by more poleward-oriented **E** vectors. The Pacific and North Atlantic jets are more zonally extended, which tends to increase extratropical cyclogenesis over the Gulf Stream (Schemm et al., 2016). To shed further light on this, we therefore explore the ENSO climatologies using troughs and ridges, which are very closely related to the upper-level eddies described above.

In Fig. 6, anomalies in the orientation of troughs and ridges are shown for ENSO-affected winter seasons based on the Oceanic Niño Index (ONI) from NOAA's Climate Prediction Center. During El Niño (Fig. 6a), troughs and ridges exhibit a stronger cyclonic orientation over the northeastern Pacific, North America and the North Atlantic. The anomalies are strongest over the northeastern Pacific and off the east coast of the US. Over the North Atlantic, the stronger-than-usual cyclonic orientation is most pronounced along 30°N, which is about the latitude of the zonally more extended jet in this region during El Niño. In contrast, during La Niña, troughs and ridges exhibit a more anticyclonic orientation over the northeastern Pacific, North America and parts of the North Atlantic. These results are in agreement with those from the previous studies that rely on the strength and orientation of the **E** vectors. The results are also in good agreement with the findings of Shapiro et al. (2001), who noted more life cycles of type LC2 (cyclonic) over the Pacific during El Niño and more life cycles of type LC1 (anticyclonic) during La Niña. The trough and ridge detection and tracking algorithm can thus enrich insights into the dynamics of upper-level eddies during ENSO-affected winters. For example, the anomalies during El Niño over the northeastern Pacific are predominantly due to more cyclonically oriented troughs, while the anomalies over the North Atlantic have a stronger signal in the ridge anomalies (not shown). The more cyclonic trough and ridge orientation over the Pacific is connected to the deepening of the Aleutian *Low* during El Niño, whereas it is a strong Aleutian *High* during La Niña, which is in agreement with more anticyclonic troughs and ridges (Mo and Livezey, 1986).

### 3.4 The North Pacific storm track in midwinter

The midwinter suppression of the North Pacific storm track intensity is another research topic that has recently received renewed attention. While the mean baroclinicity is largest during midwinter, storm track activity is reduced over the North Pacific – in contrast to the North Atlantic (Nakamura, 1992). As feature-based tracking statistics have shown, the suppression is connected to a reduced eddy intensity and upper-level eddy frequency (Penny et al., 2010), while it does not affect the frequency of surface eddies (Schemm and Schneider, 2018). Several mechanisms have been suggested to contribute to the suppression, for example a reduction in upstream seeding (Penny et al., 2010; Chang and Guo, 2011; Penny et al., 2011; Chang and Guo, 2012; Penny et al., 2013), or an increase in the jet speed with a concomitant reduction in the jet width (Harnik and Chang, 2004). Recently, however, several studies have highlighted the important role of processes internal to the North Pacific storm track, such as a reduction in the lifetime of eddies (Schemm and Schneider, 2018) and an increase in the eddy group velocity (Chang, 2001). There is growing evidence that the equatorward shift in the subtropical jet is key for understanding the suppression (Chang, 2001; Nakamura and Sampe, 2002; Yuval et al., 2018; Schemm and Rivière, 2019). When the subtropical jet over the Pacific shifts equatorward, the efficiency of synoptic systems to convert the mean baroclinicity into eddy energy is reduced due to a change in the vertical eddy structure (Chang, 2001; Schemm and Rivière, 2019). The tilt of the eddy geopotential isolines with height – which is more poleward when the eddy efficiency decreases during midwinter – is constrained by the eddy propagation direction, which is more shifted toward the equator during midwinter (Schemm and Rivière, 2019; Novak et al., 2020) – see, for example, Figs. 1 and 8 in Schemm and Rivière (2019). Equatorward jet shifts also reduce the storm-track intensity over the North Atlantic (Penny et al., 2013; Afargan and Kaspi, 2017). This is also the case in idealized aqua-planet simulations (Novak et al., 2020).

The equatorward shift of the subtropical jet over the Pacific changes the large-scale environment in which synoptic systems grow because more systems will grow on the poleward flank of the jet. The poleward flank is characterized by a large-scale cyclonically sheared environment. According to idealized wave life cycles (Thorncroft et al., 1993), we must expect more cyclonic (LC2) developments during midwinter. Indeed, troughs and ridges at the 500 hPa level exhibit a stronger cyclonic orientation in January (Fig. 7b) compared to, for example, November and April (Fig. 7a,c). Not only does the mean cyclonic orientation increase during midwinter, but also wider parts of the Pacific are affected by cyclonically oriented troughs and ridges. In particular, in January, cyclonic troughs dominate the North Pacific even equatorward of 40°N, which is not the case in November and April.

There are several observations based on trough and ridge detection that may prove useful in gaining a better understanding of the midwinter suppression. First, there is no marked reduction in the frequency of troughs and ridges during midwinter, as is the case for highpass-filtered eddies at the same height (Penny et al., 2010)[1]. A local increase in trough frequency over the eastern Pacific is observed (yellow contours in Fig. 7d) and an overall poleward movement of the ridge frequencies, which otherwise do not display a well-marked reduction. Second, the preferred trough and ridge orientation is cyclonic during midwinter, with two maxima over the eastern and western Pacific (right column in Fig. 7b), which indicates a consistent change in the character

---

[1]Note that Penny et al. (2010) use a 90-day highpass filter plus a spatial filter that admits only wavenumbers 5 to 42.

of synoptic wave life cycles during midwinter. The first maximum between 40–50°N, 165°–180°E is collocated with the region of maximum reduction in the efficiency of baroclinic growth (see Fig. 2b in Schemm and Rivière, 2019) and slightly upstream of the maximum in eddy kinetic energy (see Fig. 1 in Schemm and Schneider, 2018), which is suppressed during midwinter. Because the cyclonic orientation of troughs and ridges is typically strongest during the final stage of a synoptic wave life

cycle, a plausible – but here untested – hypothesis for the existence of the first maximum would be that synoptic cyclones generated east of Japan over the Kuroshio extension have an accelerated life cycle with a fast and intense deepening phase followed by a rapid decay, resulting in a reduced life time as found by Schemm and Schneider (2018). The enhanced lysis over the eastern Pacific is in agreement with the localized trough frequency maximum (yellow contours in Fig. 7d), because – as was shown in a previous section – maxima in the trough frequency exhibit a strong agreement with maxima in cycloysis

(compared with, for example, Fig. 5d in Hoskins and Hodges, 2002). Thus, the conversion rates and eddy kinetic energy (EKE) are reduced downstream of the maximum cyclonic orientation of the upper-level troughs compared to the shoulder seasons. Indeed, a reduction in the lifetime of synoptic systems is observed during midwinter (Schemm and Schneider, 2018). The second maximum is related to the decay of the synoptic waves at the exit of the Pacific storm track in the Bay of Alaska.

## 4  Lagrangian perspective on troughs and ridges

In this section, we show how trough and ridge detection can be used to investigate trough and ridge dynamics from a Lagrangian perspective. To this end, we couple the trough and ridge detection with a Lagrangian analysis using LAGRANTO [Wernli and Davies (1997); Sprenger and Wernli (2015)], which we use to compute air parcel trajectories from detected trough and ridge features.

The procedure is best explained with a simple example (Fig. 8). At 18:00 UTC on 18 January 2010, a trough is detected

over the Nordic Seas downstream of a mature low-pressure system south of Greenland (Fig. 8a). The trough has a mildly cyclonic orientation and will broaden during the following days. A potential research question may be: Is the formation of this downstream trough predominately driven by dry dynamics or considerably modified by diabatic processes? To answer this, air parcel trajectories are released from every grid point inside the trough feature (thin gray lines in Fig. 8a), which highlight the pathway of air parcels that constitute the trough at the 500 hPa level. The mean evolutions of pressure (Fig. 8b) and PV (Fig. 8c)

along these parcel trajectories suggest that the air is mostly advected horizontally with a minor descent of approximately 20 hPa in 24 h. Further, the diabatic modification of PV during the 24 h prior to arrival in the target region is small. The mean potential temperature (not shown) decreases during this period from 297 K to 295 K. While the mean values suggest only little change in pressure and PV, the histograms of 24 h changes in pressure and PV show that around 15 % of the air descends during the 24 h period by up to 60–80 hPa (Fig. 8d), and that a small fraction of less than 5 % decrease their PV by more than 0.4 pvu[2] (Fig. 8e).

Thus, the dynamics underlying the formation of this specific downstream trough at this stage of its life cycle are mostly dry, and therefore can be approximated by the traditional paradigm of downstream development (Simmons and Hoskins, 1979;

---

[2]$1 \, \text{pvu} = 10^{-6} \, \text{m}^2 \, \text{s}^{-1} \, \text{K} \, \text{kg}^{-1}$

Orlanski and Chang, 1993; Simmons, 1994; Papritz and Schemm, 2013). However, for a small embedded fraction of air, much stronger descent and diabatic PV modifictation than indicated by the mean are observed.

Next, 24 h backward trajectories are released from all detected troughs and ridges over the North Atlantic (60°–0°W, 20–70°N) in one winter and one summer month: January 2010, when more than 25,000 parcel trajectories are released; and July 2010 with more than 16,000 parcel trajectories. The binned 24 h changes for pressure, potential vorticity and potential temperature are presented in Fig. 9. During January 2010, the distribution of 24 h pressure changes is centered between 0–100 hPa (corresponding to a weak descent) and is highly skewed toward negative values (corresponding to a strong ascent) of up to -500 hPa, in contrast to positive values which do not exceed 250 hPa. This phenomenon reflects the nature of the air mass motion inside troughs in which equatorward-moving air mildly descends due to the quasi-isentropic downglide, while in some cases vigorous vertical motions occur – in agreement with the long tail toward positive values of the potential temperature distribution (Fig. 9e,f). We refer to the descending motion as quasi-isentropic, because for most trajectories a mild decrease in potential temperature between -2 K and 0 K is found, likely resulting from radiative cooling. Most of these air parcels are associated with only small changes in pressure. The overall asymmetry in the pressure change distribution ultimately relates to the occurrence of upward moist convection and the impact of condensation on the vertical motion in the atmosphere. This skewness is thus a general feature of extratropical cyclones in a moist atmosphere (O'Gorman, 2011). The skewness and asymmetry is assumed to increase in a warmer climate and with increasing moisture content (Booth et al., 2015; Tamarin-Brodsky and Hadas, 2019; Sinclair et al., 2020). The trajectories associated with strong vertical motion all originate in the boundary layer and first increase and later decrease their potential vorticity (not shown), which is characteristic of a warm conveyor belt ascending ahead of the trough axis (e.g., Wernli and Davies, 1997). These cases exhibit a marked increase in potential temperature, which reflects the cross-isentropic motion. For the few cases with a strong decrease larger than 200 Pa in 24 h, we find that the vast majority of the parcels – originating between 200–300 hPa – decrease their potential vorticity. Only a small fraction of approximately 1% starts with stratospheric values (>2 pvu), but we assume that this number increases for 300 hPa troughs. Finally, we note that the 24 h pressure change for July is even more confined to values between 0–100 hPa, while the 24 h change in potential vorticity is more confined between -0.2 and 0.2 pvu, which is a result that reflects the overall more intense cyclone and associated trough-ridge development and vertical motion during winter.

For the 500 hPa ridges, the histogram of changes in pressure along the flow of 24 h backward parcel trajectories is centered between -100 and 0 hPa, again with a longer tail toward negative values (Fig. 10). It is therefore shifted more toward negative values than the trough histogram, which reflects the quasi-isentropic upglide of air moving poleward and upward in a ridge. The pressure change in July is again more shifted toward negative values (ascent) than the trough histogram but is centered around -50 and 50 hPa. If our interpretation is correct, the histograms are dominated by the almost isentropic motion of air inside troughs and ridges, and the reduced pressure change along the flow in July relative to January reflects the reduced baroclinicity or isentropic tilt in summer compared to winter, in combination with a reduced wind speed according to the thermal wind relationship. It is, however, important to note that the histograms of potential temperature and vorticity changes show that the motion is not fully dry adiabatic. The histograms of potential vorticity changes are symmetrically distributed around zero with

outliers on both sides of the spectrum, while the distribution of the 24 h potential temperature changes are centered around weak negative values (between -2 and 0 K), which is similar to the troughs.

## 5 Conclusions

### 5.1 Summary

In this paper, a novel method to detect the full life cycle of upper-level troughs and ridges is presented, which is computationally efficient and closes an existing gap between tools that focus either on the wave breaking phase or on the initiation phase of upper-level Rossby waves. The presented approach is based on the curvature of the geopotential isolines at a given isobaric surface. Grid points above a predefined curvature value are grouped into two-dimensional features and labeled as trough or ridge features. The algorithm further categorizes the corresponding trough and ridge axes as line objects and identifies their horizontal orientation. It also performs a spatio-temporal tracking of the identified features. The detection of the trough and ridge axes allows for the identification of pre- and post-trough/ridge sectors, respectively. This facilitates a variety of new applications relating troughs and ridges to other meteorological variables such as cloud types, precipitation, warm conveyor belts or jet streams. The detection of the horizontal orientation allows for the grouping of troughs and ridges into cyclonic and anticyclonic categories. The tracking enables determining the maturity of the troughs and ridges and the orientation changes during the life cycle. Finally, the algorithm separates closed from open troughs and ridges. The former are associated with one or more closed geopotential isolines and therefore related to upper-level lows or highs, which allows them to be connected with surface low-pressure systems.

The second focus of this paper is on illustrating the detection using a case study and several climatological applications. These serve as stepping stones for future research. In general, the detection frequencies of both troughs and ridges are remarkably stationary, but regionally the picture can be complex. For example, during winter, troughs are most frequently detected downstream of the Rocky Mountains and downstream of the Atlai, but they are much less frequent downstream of Greenland – where, by contrast, the ridge detection frequency is high. Thus, troughs are more transient or smaller near Greenland. Troughs are also frequent at the end of the Pacific storm track and at the end of the Atlantic storm track during winter, where the frequency maximum of incipient troughs is high over the British Isles extending over eastern Europe. Rather, a frequency maximum is located over Eastern Europe – which is in contrast to summer, when there is a clear maximum at the end of the Atlantic storm track off the Iberian Peninsula (Fig. 4 and Fig. 5).

The horizontal orientation of troughs and ridges displays a strong meridional and longitudinal dependency. In general, troughs and ridges are oriented cyclonically over the main oceanic storm tracks but anticyclonically over land. The cyclonic orientation increases poleward, while the anticyclonic orientation increases equatorward. In agreement with earlier findings – based on traditional means to quantify the behavior of upper-level eddies, such as **E** vectors – troughs and ridges are more cyclonically oriented over North America and the North Atlantic during El Niño-affected winter seasons. By contrast, these areas are more anticyclonically oriented during La Niña-affected winter seasons. Trough and ridge detection thus provides

complementary insights into the change in the dynamics of the large-scale upper-level flow in response to external climate drivers such as ENSO.

The orientation of trough and ridge axes also exhibits a seasonal cycle, for example during midwinter over the North Pacific when the Pacific jet moves equatorward. The subtropical jet creates a large-scale cyclonically sheared environment over the Pacific. Consequently, the troughs and ridges during midwinter are mostly oriented cyclonically. The trough orientations in particular exhibit a marked increase in cyclonic orientation, suggesting a potentially accelerated cyclone life cycle, which is in agreement with an emerging trough frequency maximum over the northeastern Pacific. The latter suggests enhanced cycloysis and wave breaking in this region. These applications serve as an indication for the usefulness of the novel detection method and complementary insights that can be gained from it.

Finally, we explored the possibility of utilizing the detected troughs and ridges as starting points for air parcel trajectories. Twenty-four-hour backward trajectories are released for two months (January and July 2010) from all the trough and ridge objects. For troughs, the distribution of 24 h pressure changes along the flow is centered between 0 and 100 hPa, which reflects the near-isentropic downglide of equatorward moving air in a trough, with a long tail toward negative values (strong ascent). For ridges, the distribution is consequently centered between -100 and 0 hPa, which reflects the near-isentropic upglide of poleward moving air in a ridge. Again, the distribution has a long tail toward negative values. During summer, the tails of both distributions are reduced and strongly centered around small pressure changes of 50 hPa. If the distributions reflect mainly the near-isentropic motion of air masses in trough and ridge regions, then the strong confinement of the summer distributions near small pressure changes is due to the reduced baroclinicity. The corresponding distribution of PV changes is centered around zero, with similar tails toward positive and negative values. Finally, the distribution of potential temperature changes is centered around negative values (between -2 and $0 \, \mathrm{K} \, \mathrm{d}^{-1}$), with a long tail toward positive values. While the long tail reflects the heating during rapid cross-isentropic ascent, the mean and median indicate a mild loss potentially due to radiative cooling. Consequently, the mean motion of air in a trough and ridge is only close to isentropic, with stronger positive than negative outliers (Figs. 8e and 9e).

The presented tool offers new possibilities for studying the dynamics of upper-level wave development and associated meteorological conditions at the surface or at the jet stream level. Their application to climate data or century-long reanalysis data will allow for the analysis of long-term trends, extreme seasons or decadal variability in the frequency and orientation of troughs and ridges. The combination of the tool with Lagrangian diagnostics will allow for the development of a new three-dimensional perspective on trough and ridge development and for a quantification of the role of moist processes in shaping their life cycle.

## 5.2   Caveats and Outlook

The presented approach to detect upper-level troughs and ridges successfully identifies all principal tracks shown in previous studies (e.g., Hoskins and Hodges, 2002) with the exception of the southern branch over Eurasia (Chang and Yu, 1999; Chang, 2005). While the method is in general capable to identify troughs and ridges along the southern branch (an example is shown in the response to the reviewers' comment, which is available in the interactive discussion forum of this publication), this seem

to happen less frequently as one might expect. The underlying reasons can be twofold. Previous studies have in different ways pre-processed the input data by removing the stationary waves or via spatial or temporal filters. Another reason could be the latitudinal dependency of the curvature of the climatological mean geopotential isolines. Near 25°N, the mean curvature is 0.07 degrees per km, while it is 1 degree per km near 55°N. The used global threshold of 0.05 degrees per km is exceeded

5    more often at higher latitudes compared to lower latitudes. In an upcoming version of the algorithm that uses ERA5 as input data, we will provide the possibility of using a threshold that varies with latitude.

*Code and data availability.*  The monthly mean data will be made available at http://eraiclim.ethz.ch/. Higher resolution data and the Fortran and Python codes can be provided upon request.

*Author contributions.*  SeS and MS designed jointly the detection strategy. MS developed the algorithm. SR developed the tracking. SeS

10    performed the analyses. SeS and MS contributed equally to the interpretation and discussion of the results. All authors helped to write the paper.

*Competing interests.*  The authors declare no competing interests.

*Acknowledgements.*  We acknowledge the comments from three reviewers, in particular the discussion surrounding the southern branch over Eurasia, which helped to improve our manuscript.

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

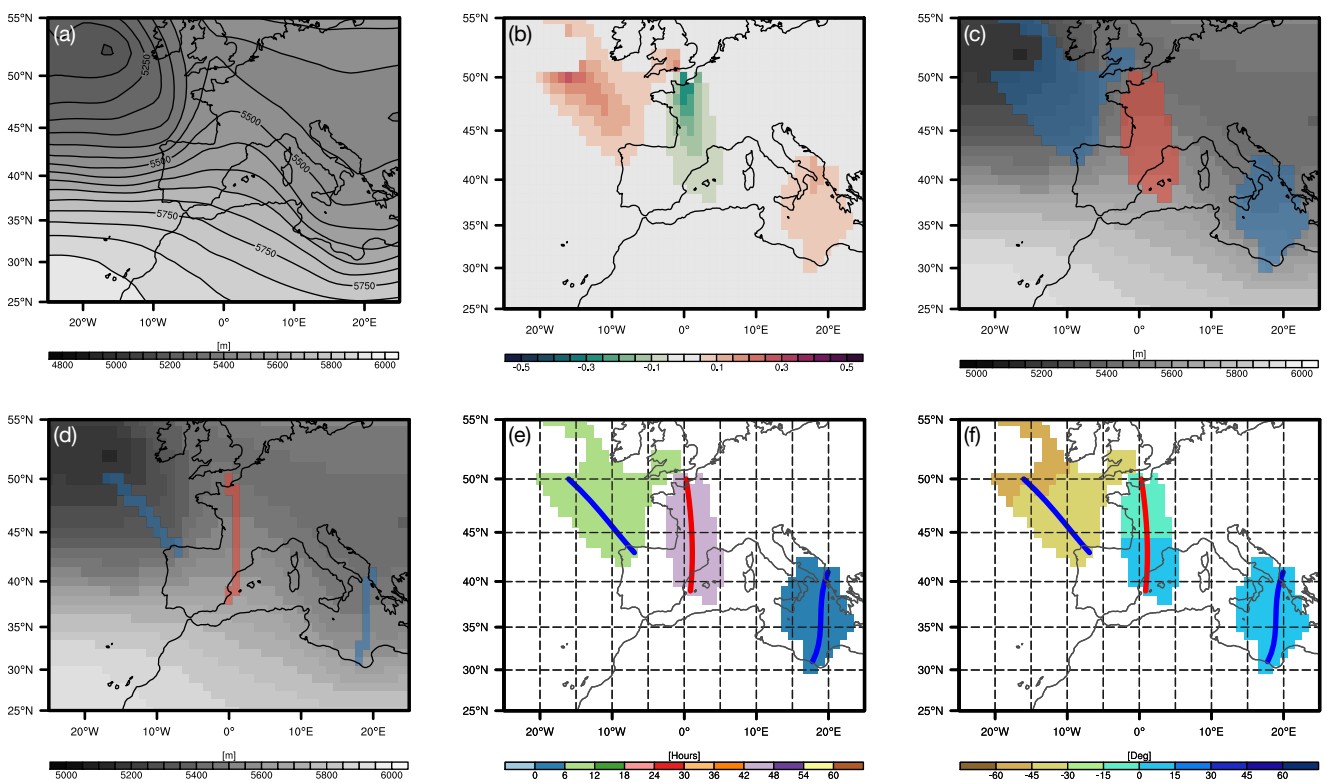

**Figure 1.** (a) 500 hPa geopotential height (m) at 12 UTC on 12 January 2010; (b) curvature of the geopotential isolines (units: degrees per kilometer) on the $1° \times 1°$ input grid at every grid point where the curvature is larger than 0.05 degrees per km; (c) 2D trough (blue) and ridge (red) masks and geopotential height (gray shading) at every grid point; (d) corresponding trough (blue) and ridge (red) axes; (e) cubic-spline interpolated trough and ridge axes and age (color; in hours) of the trough and ridge features; (f) horizontal orientation of the trough and ridge object (in degrees; angle relative to a north-south meridian and estimated from the corresponding axis).

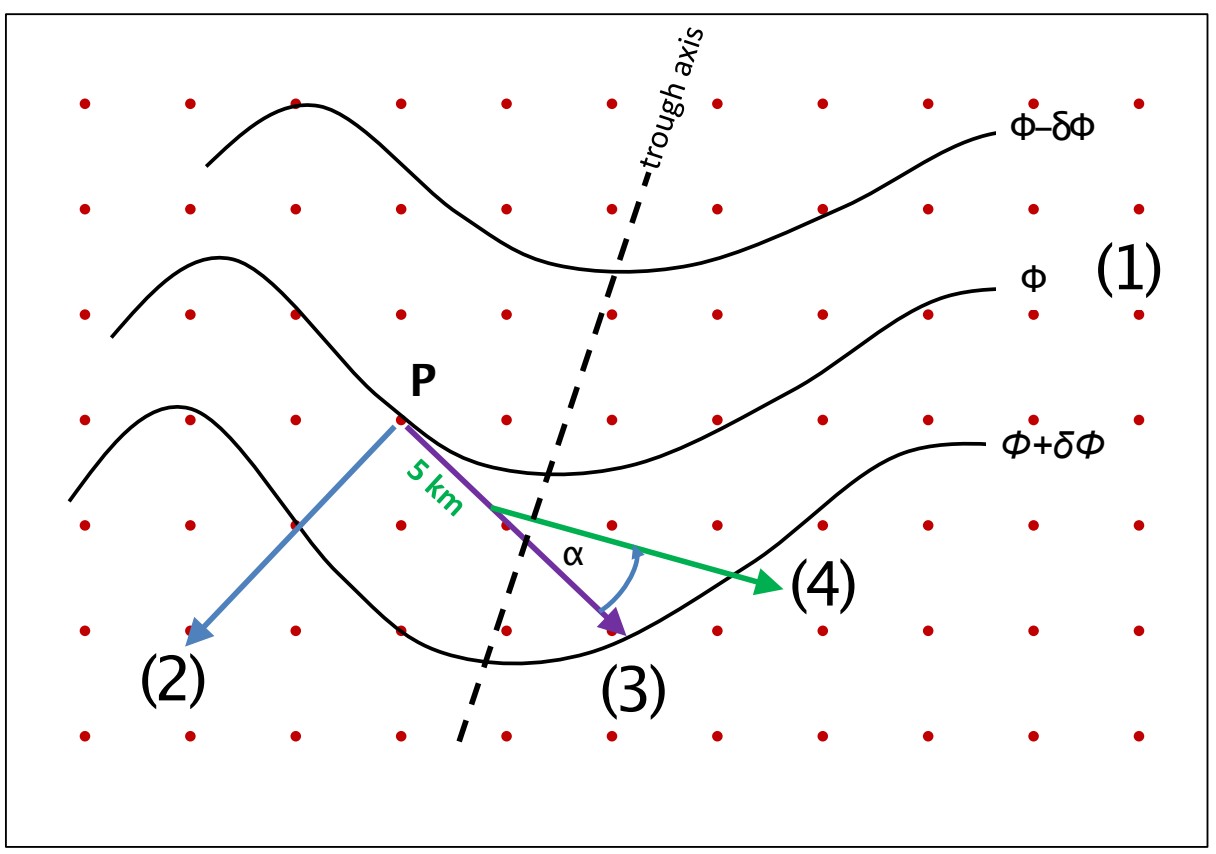

**Figure 2.** Schematic showing the different steps in determining the local curvature of the geopotential height isolines. The steps are: (1) geopotential height isolines (black contours) at every grid values (red points); (2) local gradient vector (blue) of the geopotential height at a specific grid point P; (3) 90-degree rotated vector (purple) to get a tangent vector parallel to the geopotential height isolines; (4) a 5 km step along the geopotential height isolines in the direction of vector (3) and a new tangent vector (green) derived at the new position; (5) calculating the angle $\alpha$ between the vectors (3) and (4). For further details, see text.

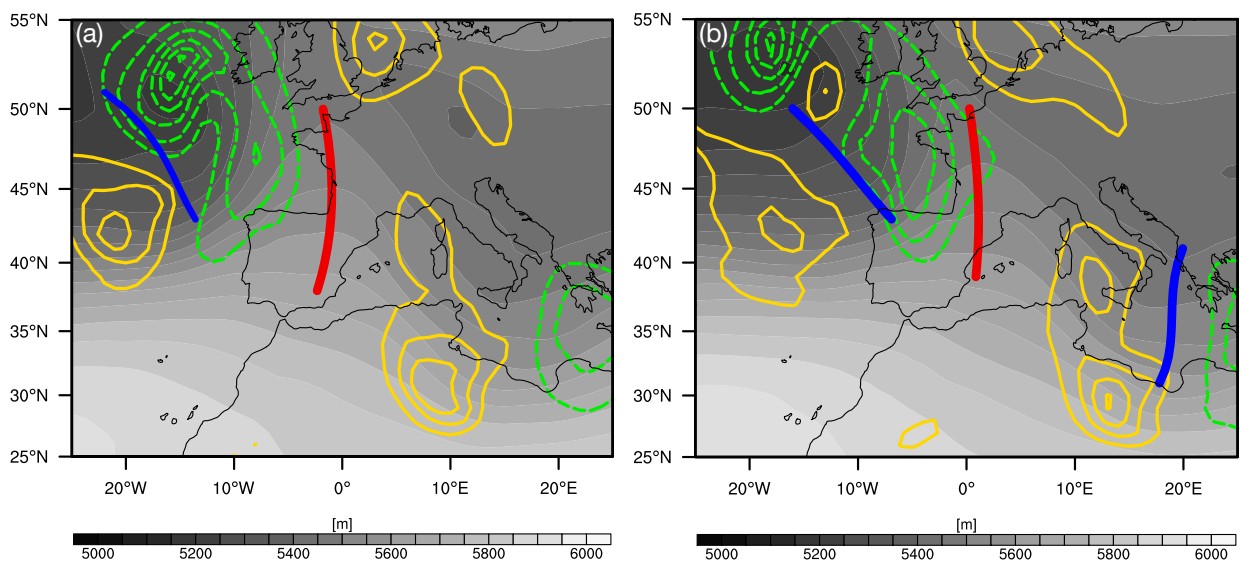

**Figure 3.** 500 hPa trough (blue) and ridge (red) axes at (a) 06 UTC and (b) 12 UTC on 12 January 2010, quasi-geostrophic omega (from 0.1 to 1 in steps of 0.1 m s$^{-1}$; positive values in yellow indicate descent and negative values in green indicate ascent) and geopotential height (m; gray shading).

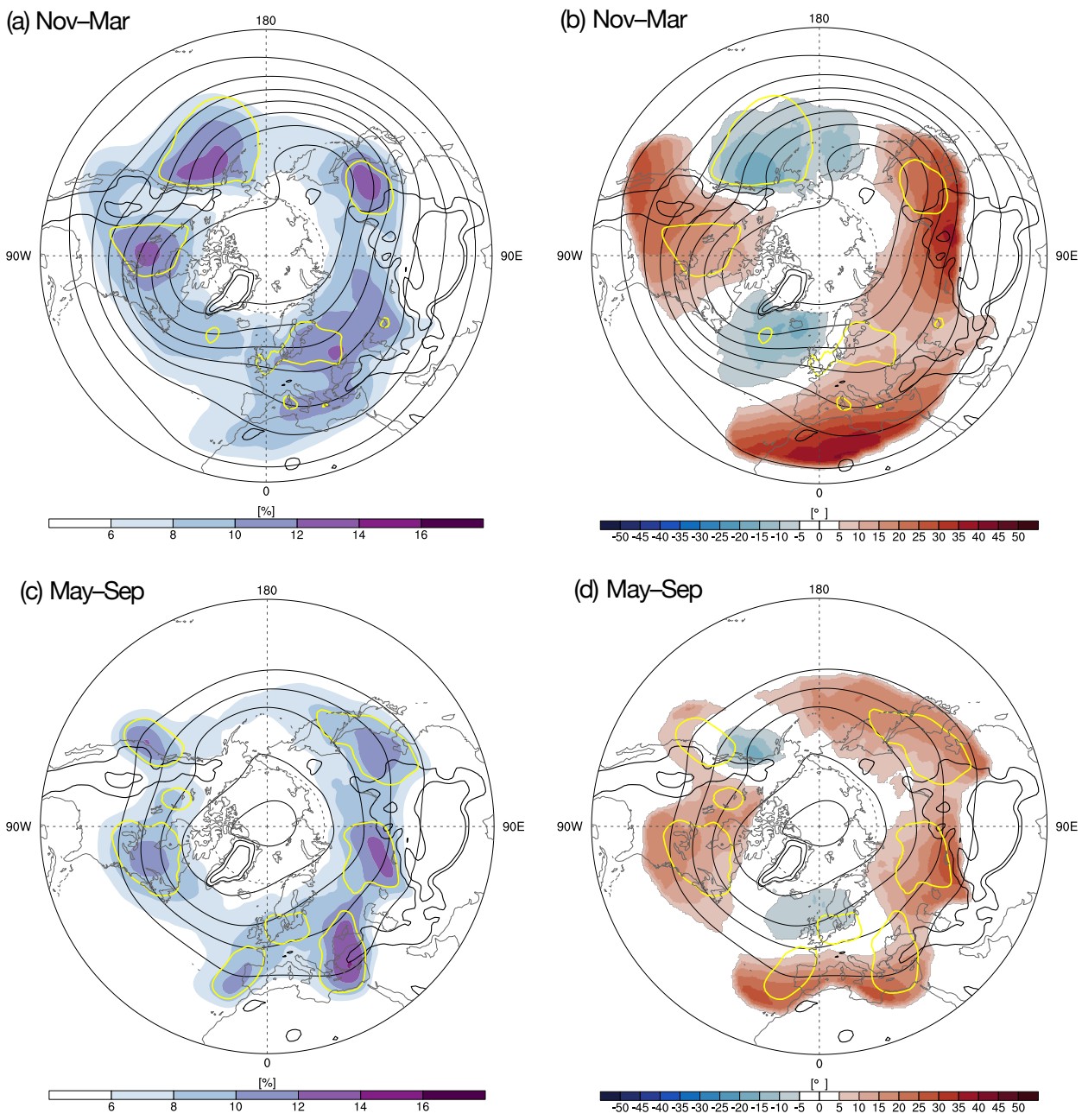

**Figure 4.** Left column: Seasonal climatologies of the trough detection frequencies (color shading; units: %) for the cold (a; Nov–Mar) and warm (c; May–Sep) seasons. Additional contours show selected frequencies of troughs with an age between 0 and 24 hours (yellow). Right column: Seasonal climatologies of the corresponding horizontal trough orientation on the 500 hPa level (color shading; units: degrees) for the cold (b; Nov–Mar) and warm (d; May–Sep) seasons. Positive values indicate anticyclonically oriented troughs. Additionally, the yellow 0–24-hour age contour, similar to that in (a, c), is shown for reference. The 1,500-m height contour (black) is shown in all the panels.

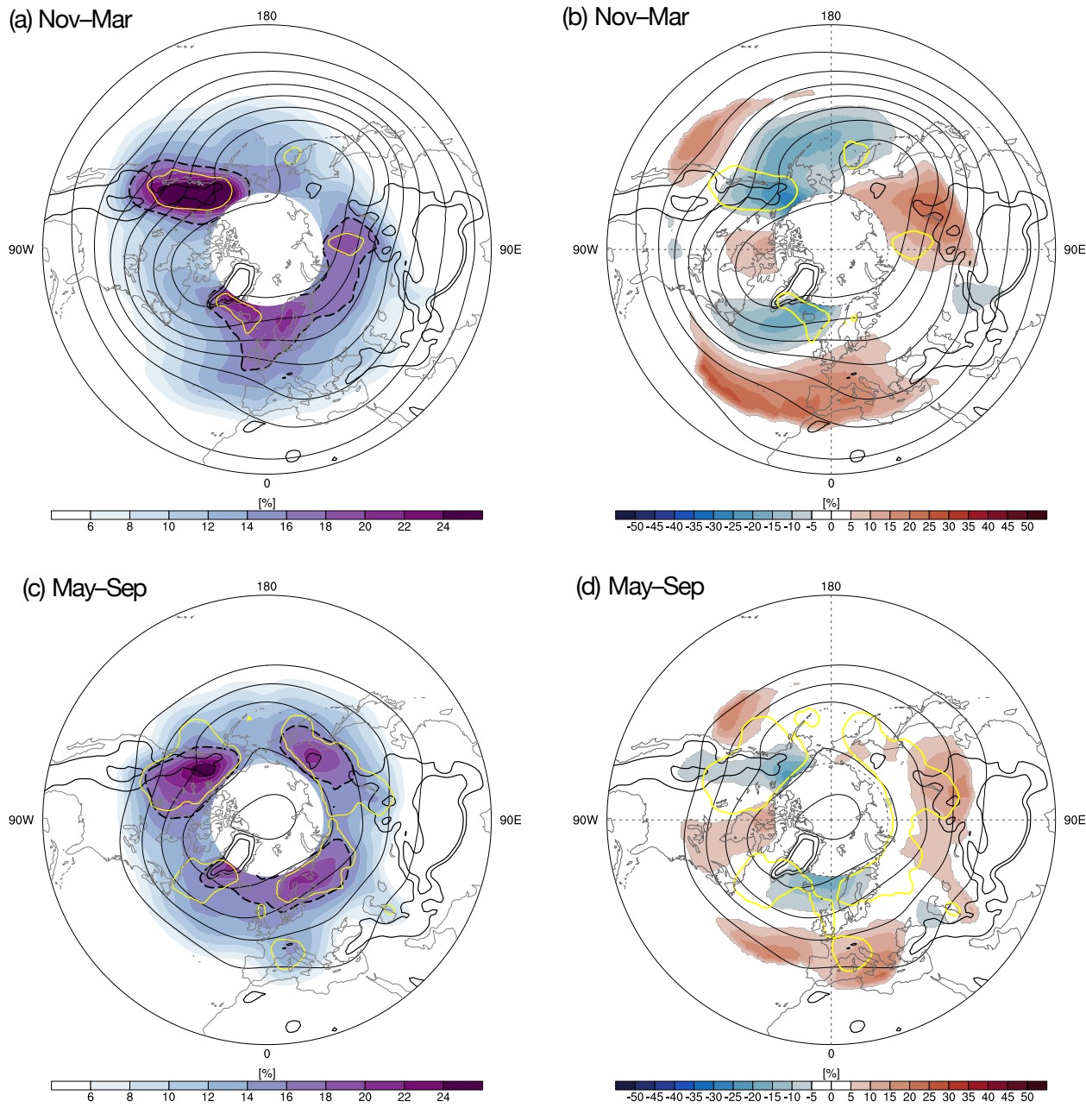

**Figure 5.** Similar to Fig. 4 but for ridges. The dashed contour indicates the region where the ridge frequency is above the maximum global trough detection frequency ($\approx 16\,\%$).

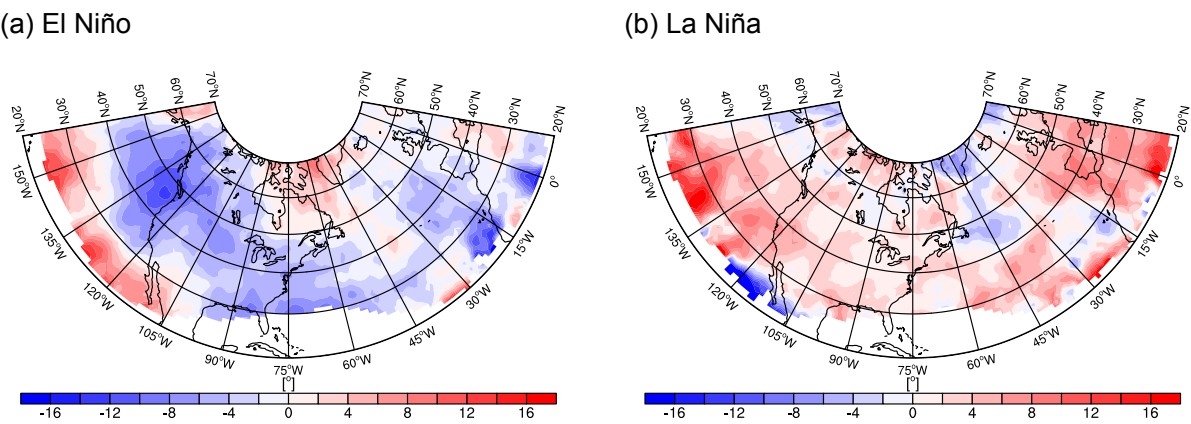

**Figure 6.** Averaged orientation anomalies (units: degrees) of 500 hPa troughs and ridges during (a) El Niño- and (b) La Niña-affected winter seasons. (Positive) Negative anomalies indicate more (anti)cyclonically oriented troughs and ridges compared to the seasonal climatology. Regions where the detection frequency is below 2 % are excluded.

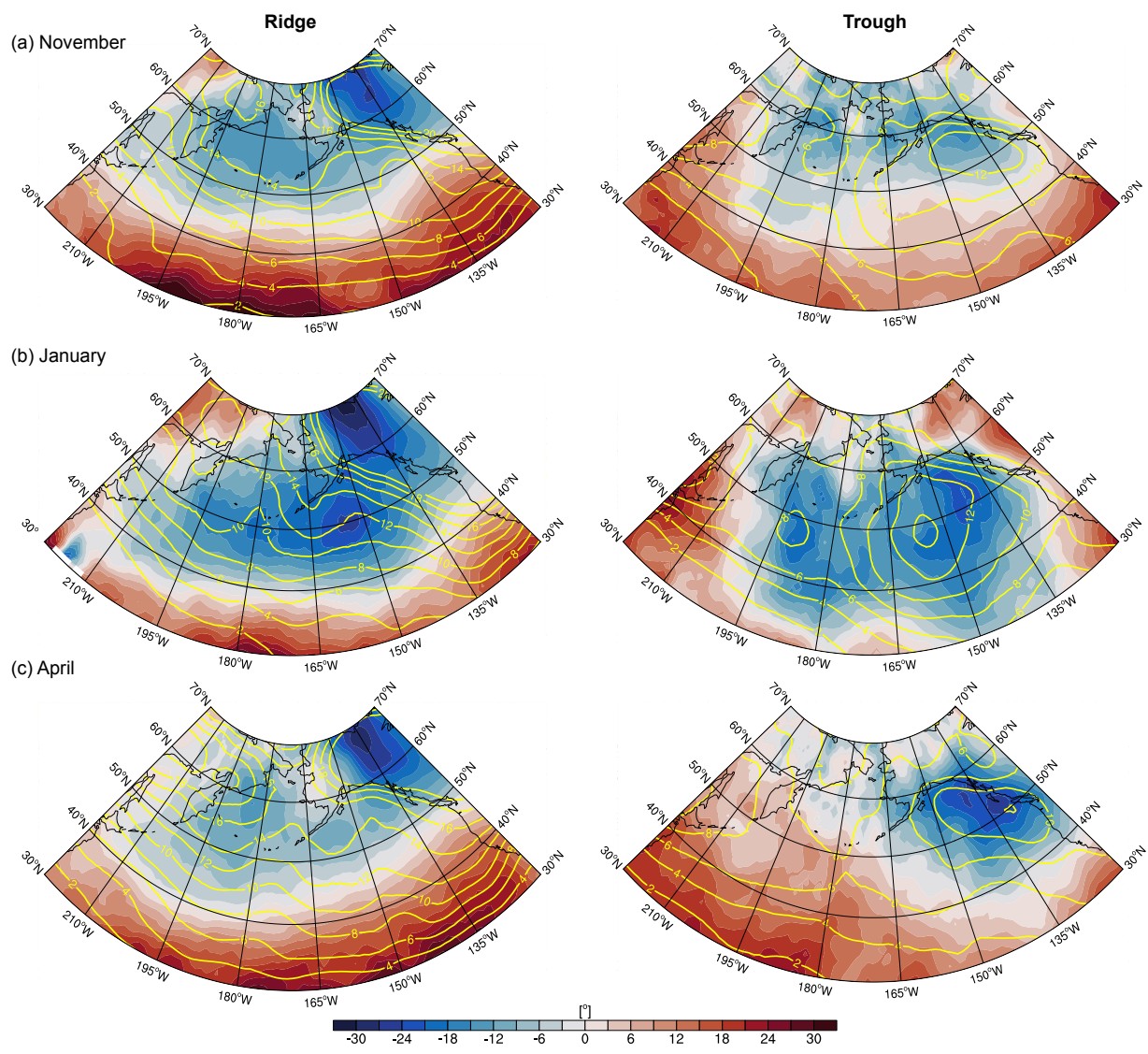

**Figure 7.** Monthly mean orientation (color shading; degrees) and frequency (yellow contours, from 2 to 14 % in steps of 2 %) of 500 hPa ridges and troughs during (a) October, (b) January and (c) April over the North Pacific.

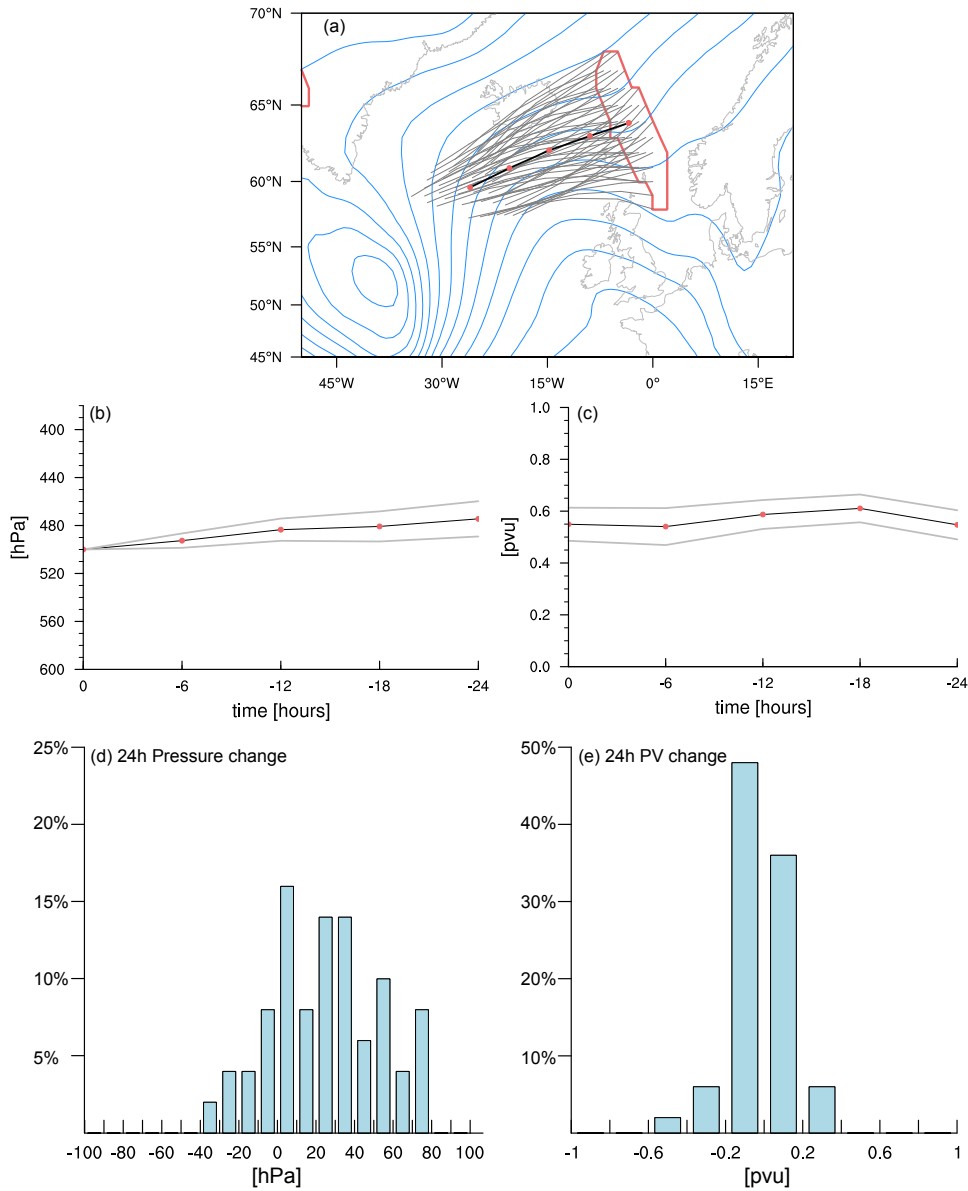

**Figure 8.** (a) Twenty-four-hour backward trajectories (gray lines) released on 18:00 UTC 18 January 2010 from a trough feature (red contours) at 500 hPa. The black contour indicates the average over all the trajectories. Red dots indicate 6-hourly time steps. Geopotential heights are shown as blue contours (5000 to 5600 m in steps of 50 m). (b) Mean pressure evolution (black; hPa) along the backward trajectories shown in (a). The standard deviation across the sample is indicated by thin gray lines. Red dots denote 6-hour steps. (c) Similar to (b) but for the potential vorticity (pvu). (d,e) Histograms of the 24-hour changes in pressure (hPa day$^{-1}$) and potential vorticity (pvu day$^{-1}$)

.

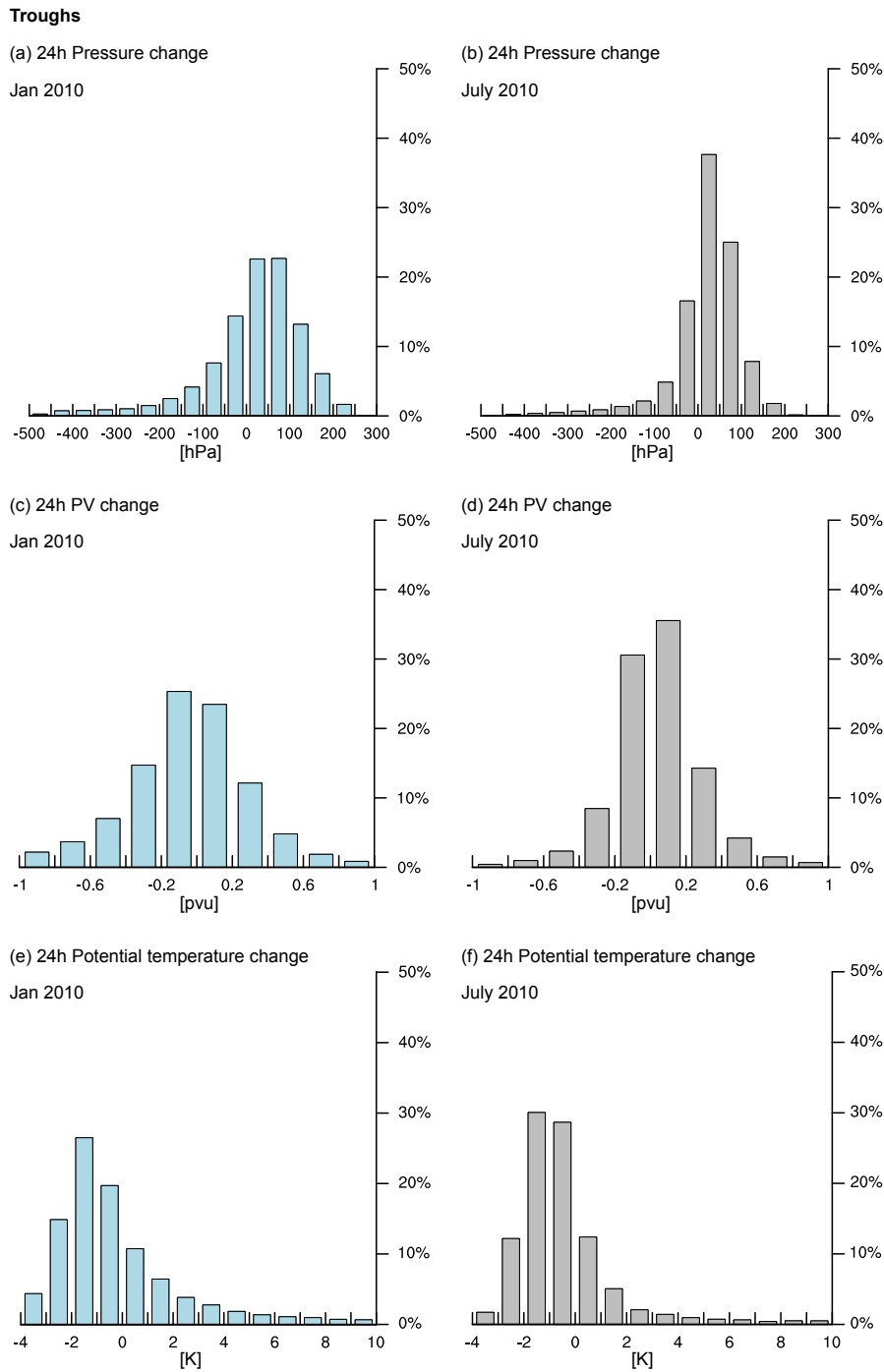

**Figure 9.** Histograms of the change (a, b) in pressure (hPa day$^{-1}$), (c, d) the potential vorticity (pvu day$^{-1}$) and (e, f) potential temperature (K day$^{-1}$) along the air parcel trajectories released from all troughs detected at the 500 hPa level during (a, c) January and (b, d) July 2010.

**Ridges**

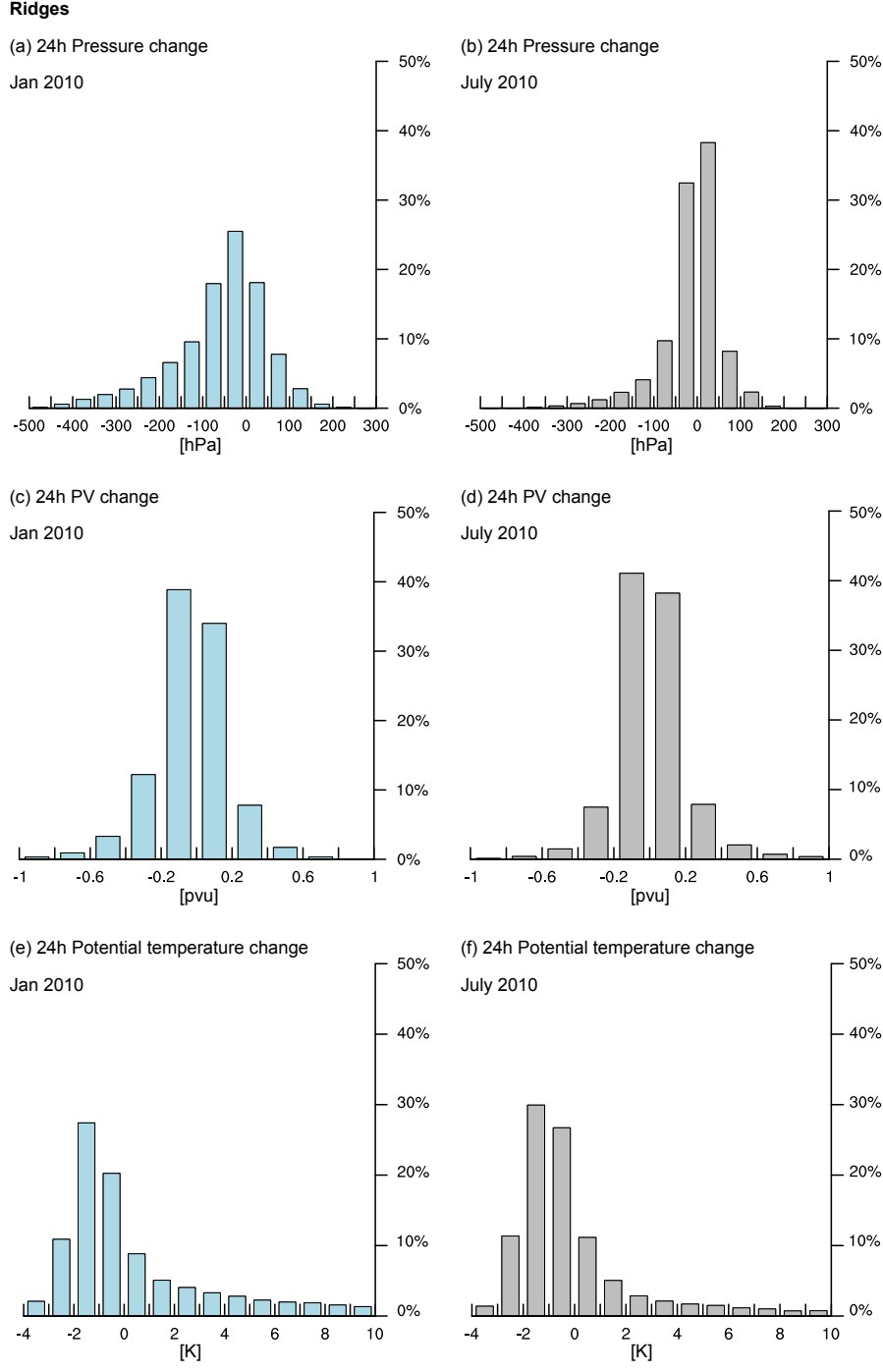

**Figure 10.** Similar to Fig. 9 but for 500 hPa ridges.