# Peer review of "The Life Cycle of Upper-Level Troughs and Ridges: A Novel Detection Method, Climatologies and Lagrangian Characteristics"

_Weather and Climate Dynamics, 2020_

## Referee Comment (RC1) · Anonymous Referee #1 · 17 Apr 2020

In this paper, the authors developed a new method to track trough and ridges. They defined ridge and trough axes using curvature, and tracked ridges and troughs and record both the areas covered by the objects (based on a threshold of curvature) as well as the orientation of the axes. In so doing their method provides additional information on top of those provided by previous studies. They applied this method to compile climatological statistics for the Northern Hemisphere extended cold and warm seasons, and examined ENSO impacts on trough and ridge orientations as well as the winter seasonal cycle in the Pacific. Finally they used Lagrangian back trajectory from within

trough and ridge areas to examine the behaviour of air parcels making up the trough and ridges over the previous 24 hours. Overall the results are interesting and the method is new and can provide new insights over previous methods. However, I do have concerns which should be addressed before this paper can be accepted for final publication.

Major concerns: 1) My biggest concern is with the authors' choice of 500 hPa geopotential height (Z500) as the variable for analysis of so called "upper-level" troughs and ridges. As far as I can see, no justification is provided for the choice of this variable. Historically Z500 was widely used in the 1970s and 1980s for analyses of troughs and ridges. However, since PV thinking became mature (cumulating in the seminar paper by Hoskins et al. 1985), synoptic dynamicists have generally accepted that troughs and ridges are manifestation of PV anomalies that have largest amplitudes either at the tropopause or at the surface, and in recent decades, most analyses have focused on analyzing variables either near tropopause level or near the surface (e.g. Hoskins and Hodges 2002, Fig. 1). In this paper, the authors pick the mid-level (500 hPa) for analyses of upper level troughs. Can the authors please provide justification why they pick a mid-troposphere variable instead of an upper troposphere varable to analyze?

References:

Hoskins et al., 1985, QJRMS, 111, 877

Hoskins and Hodges, 2002, JAS, 59, 1041

2) Also why use curvature of geopotential height contours? Why not use relative vorticity or PV instead? Given that dynamically, we can easily write equations for vorticity or PV tendency while it is difficult to write an equation governing the tendency of curvature of Z500 contours, what are the advantages for picking such a variable to analyze? Can't trough/ridge axes be defined based on vorticity or PV?

3) A potential issue is that results can be affected by high terrain, e.g. the Tibetan

Plateau. Previous studies (e.g. Chang and Yu 1999, Hoskins and Hodges 2002 (HH02 hereafter); Hakim 2003) have shown that there are upper level waves that propagate along the subtropical jet in winter near the southern edge of the Tibetan Plateau. These waves are clearly missing from this study (Fig. 3a and 4a). These waves are potentially important in understanding Pacific cyclongenesis (e.g. Chang 2005) and the mid-winter suppression (Nakamura and Sampe 2002).

References:

Chang and Yu 1999, JAS, 56, 1708

Hakim, 2003, Mon Wea Rev, 131, 2824

Chang, 2005, Mon Wea Rev, 133, 1998

Nakamura and Sampe, 2002, GRL, 29, 1761

4) While the authors discussed some consistencies with previous studies for summer climatology (section 3.2), for the much more researched winter, they didn't provide much comparisons with previous studies. More reference to previous studies should be made in section 3.1. There are some differences that could potentially be due to differences in methodologies. For example, comparing their Fig. 3a to Fig. 11d of HH02, it is apparent that the authors' results are quite a bit further north than those presented in HH02. Also, the authors' results (Fig. 4a) show nearly absence of ridges to the south of Japan, while HH02 shows that 250 hPa Z positive tracks have a maximum south of Japan over western Pacific. Can the authors please explain what might be the reason(s) behind such differences? In any case, comparisons with results from previous studies should be discussed more.

5) In my opinion, the methodology is not described in sufficient details. How the tracking was done was not really presented and readers are referred to a Ph.D. dissertation (221 pages long). This seems to me an important step since one of the parameters the authors emphasized is age of the system which depends critically on the tracking algorithm. Details of how tracking is done should be presented - perhaps as an appendix or supplemetal material.

6) As the authors mentioned, there are always rather arbitrary cutoff values in any objective algorithms. For this algorithm, they mentioned three - the minimum curvature, the size of the trough/ridge objects, and the length of the axis. How sensitive are the results of these choices? How are these choices set? For example, in the example shown in Fig. 2, the trough over the Mediterranean in panel 2a looks to me a very legitimate trough, but it was not identified. I would guess that even 6 hours earlier that trough was already apparent. So the question is: what is the justification for the authors to pick those particular cutoff values? This is important for genesis and lysis, and for trough ages and lifetimes. In my opinion, sensitivity to these cutoff parameters should be explored and discussed.

7) Many objective tracking algorithms (e.g. HH02) employ other cutoff values, like feature lifetime and distance travelled. As far as I can see no such cutoffs are employed here. Could the statistics be heavily contaminated by very transient short-lived features like heat lows, or quasi-stationary climatological features?

8) Some of the statistics shown are not clearly defined. For example, what does trough detection frequency mean? Is that defined based on trough axis or trough objects? Also, what does the "selected frequencies of troughs with an age between 0 and 24 hours (yellow)" in the figure caption of Fig. 3 mean? Without clear definitions of these parameters it is difficult for readers to interpret these figures. Mathematical definition should be provided.

9) For the trough age, it is surprising to see that eastern Pacific/Gulf of Alaska is apparently a region where there is high frequency of 0-24 hour troughs (Fig. 3a). I would imagine only regions where there is frequent trough genesis would be highlighted, and that is certainly not a region that is associated with frequent trough genesis. As far as I can see the authors did not mention that region. Can the authors please discuss that?

10) Apart from the Gulf of Alaska, it seems that "young" troughs and ridges are most frequent over regions with high trough and ridge frequencies (Fig. 3 and 4). Those regions are expected to be regions where troughs and ridges are quasi-stationary and hence presumably "old" rather than "young". Nevertheless, it depends on what the yellow contour really shows as it is not really defined, but this point should be further discussed.

11) In the Lagrangian analysis, one of the authors' goals is to quantify the diabatic impact (p. 14, line 8). My question is whether 500 hPa level is the best level to do that? It is still within the cloud layer at a location where strong heating is still going on. Wouldn't the full impact of diabatic heating be more clear at the tropopause level?

Other comments: i) p. 4, line 21: 20S-70N: The authors discussed that they focus on mid-latitude troughs/ridges, but apparently did the analysis in the tropics also. In the tropics, wouldn't the fields be very noisy given the weak geopotential gradient? Why is it necessary to perform the analysis down to 20S?

ii) p. 6, lines 4-5: I'm not sure I understand what the authors meant by the sentence "Interestingly here it is a trough over the North Atlantic and the ridge downstream already exists for a longer time period than the up- and downstream troughs". Which troughs and ridges are they referring to in these descriptions?

iii) While the trough/ridge tilt is indeed closely related to the E-vectors, fundamentally tilted troughs/ridges are related to eddy momentum fluxes (which make up the E-vectors) and have been discussed as early as Jeffries (1926) and Starr (1948).

iv) p.8 line 25: Here the authors write that there is a second maximum near Lake Baikal, which to me is not really accurate since Lake Baikal apparently is located between two relative maxma, one west of 90E and he other over eastern Siberia, northeastern China.

v) Fig. 5: Are the anomalies shown statistically significant? Some of the anomalies

seem rather small to me.

vi) p. 10, lines 22-23, "more systems will nnow grow on the poleward flank of the jet" this should be quantified or reference cited. Both the jet and storm track shifts equatorward during mid-winter so it is not entirely clear that more systems grow on the poleward flank of the jet. Perhaps a histogram showing the frequency as a functio of tilt would quantify this.

vii) p. 10, lines 25-26: Same comment as above.

viii) p. 10, lines 31-32: "there is no marked reduction in the number of troughs and ridges" This is surprising given the strong decrease in number of troughs found in Penny et al. Can the authors show the results and provide some explanation on why there is such disagreement between their results and those of previous results?

ix) p. 11, line 6: "a fast and intense deepening phase followed by a rapid decay". The familiar LC2 lifecycle of Thorncroft et al (1993) shows a cyclonic lifecycle that decays very slowly. Perhaps the authors should provide references that show rapid decay of cyclonic lifecycles.

x) p. 12, line 14: Should refer to Fig. 9 here.

xi) p. 12, lines 19-20: The windspeed in summer is also weaker so even with the same tilt the vertical motion would still be reduced. This again is related to weaker baroclinicity through the thermal wind relation but still this should be mentioned. Both reduced tilt and reduced wind speed can contribute to decrease in vertical displacement.

---

## Referee Comment (RC2) · Irina Rudeva (Referee) · 28 Apr 2020

The paper presents a novel algorithm to detect troughs and ridges in the troposphere. The method is applied to 500 hPa ERA-interim data to show climatologies of these events and establish a link between their orientation to the ENSO. The approach offered in the paper is interesting and opens possibilities for future research on the role of trough/ridges in jet position, cyclogenesis, precipitation, etc. Overall, the method is new and results are valuable. I believe this paper may be accepted for publication after some revision.

**Major:**

My major comment has already been picked up by another reviewer: I would also like to see justification of the geopotential level used to identify troughs/ridges and, possibly, how results differ between levels. I can see a point in using an Hgt level as opposed to, e.g., PV, as Hgt levels are readily available in most model outputs, making this approach useful for analysis of, e.g., CMIP6 models. However, troughs/ridges are often seen as upper-level phenomena, while 500 hPa lies in the middle of the troposphere.

Other comments:

Abstract: It is said that tracking allows analysis of evolution of troughs/ridges. However, in my view, this paper does not present such analysis. Either remove this statement from the abstract (it can be kept in the description of the method) or add analysis of their life cycle.

p1, I.21-25: Add a figure showing a trough and a ridge.

p2, I.18: define E vector here

p.4, I.9-16: This part needs to be illustrated - draw a figure showing vectors discussed here

P.5, I28: give a brief description of Q vector here first, not only a reference

p.6,I22: can the absence of troughs be a result of northern boundary at 70N? Could it be that troughs extent further north and the part to the south of 70N is below the threshold on the length?

p.5, I. 32: The map at 06 UTC also suggests a significant trough in the Mediterranean, but it remains undetected. Is it due to the thresholds? In this case, some sensitivity test should be shown to justify the selection of thresholds.

p.6, l. 17: Is it frequency of trough area of trough axes?
**Discussion** paper

p. 7, I.1: To confirm that Greenland troughs are more transient can you built pdfs of trough lifetime characteristics (propagation speed and lifetime) in regions of interest (over Greenland, Pacific that is discussed later and a reference region)

p7, I28-30: Given the difference in frequency of troughs and ridges, did you think of applying different thresholds for their identification. Troughs and ridges can be seen as advection of cold and warm air and, due to the variable air density, typical curvature of hgt may vary.

p.8: Can you think of reasons why there is no cyclone maximum near the Iberian Peninsula associated with a peak in trough frequency and Rossby wave breaking? I suggest that these conditions in the upper (middle) troposphere produce cut off lows not visible on the surface.

p.8, I34: Despite equatorward shift of troughs off the West coast of the US in summer, the ridges shift poleward during summer (something that one would have expected). Can you discuss what may cause summer poleward shift of troughs in this region?

p.10, I33: The text refers to Fig 6, which shows mean orientation of ridges and troughs. This plot is supposed to illustrate the 'dominance' of those systems in the Pacific, however, I think the frequency should be shown instead.

p.11: Did you test longer backward trajectories? Do troughs/ridges have life cycles? It will be nice to see some analysis on this. Are back trajectories in fig. 7 b,c sensitive to the stage of their development?

p.12: A histogram of north/south displacements might be interesting. It is worse noticing that fig. 8a,b and e,f are in good agreement with each other, i.e. decent corresponds to a reduction in potential temperature. However, then the statement that the downward motions occur along isentropic surfaces sloping equatorward is questionable (though I like to think that the flow is along isentropes).

p.12, l2: what is the difference between a trough with closed geopotential isolines and
a cyclone at that level?

p14, l.10: Is the code available online?

Minor:

Abstract: "methods that detect the initiation phase of upper-level Rossby wave development" - did you mean Rothlisberger et al. 2016 paper or another? I could not find a paper on this in the references.

- p1, I.22: for a good reason
- p1, I.24: replace enchanted with increased
- p2, I.3: re-phrase to read better

P6,I19: change to 'Altai' (multiple times in the paper). Not sure whether this is still east Asia. Perhaps, Far East is better.

P.6,I20: remove 'slightly'

Fig. 3, 4: On my screen continents are barely visible. As for elevated areas, I suggest either marking them with a different colour or hatching.

p.11, last line: I am not sure why you say 'pushing air', I guess it is being pushed. Perhaps 'moving air' is better.

p.12, l6: remove 'most'

p.12, I3: change 'ore' to 'or'

---

## Referee Comment (RC3) · Anonymous Referee #3 · 10 May 2020

This is an interesting and well written paper. It describes a new method for tracking troughs and ridges based on curvature of geopotential height contours. The tracking algorithm features important tools, allowing to find the major axis and lifetime of the systems. After reviewing the tracking algorithm the authors present climatology of the detection frequency and axis orientation. The authors then use the unique capabilities of the tracking algorithm to analyse two important open questions: the effect of ENSO on extratropical weather systems and the Pacific midwinter minimum. Finally, the authors demonstrate the usefulness of the described tracking tool in a Lagrangian

analysis, by tracking air parcels from trough and ridges identified regions using the LA-GRANTO algorithm. The described tool is very useful and definitely worth publication following some modifications as described below.

Major comments:

1. As one of the most important skills of this tracking algorithm is finding the axis of the system, and a lot of the analysis is based on it, more effort should be invested in explaining the notation of cyclonic and anticyclonic tilt, what are the factors controlling the axis orientation and the effect of the axis orientation on the life cycle of the system and it's interaction with lower-level systems. A cartoon demonstrating these concepts might be helpful.

2. In the introduction, more attention should be given to explain the open questions addressed in this work (ENSO, midwinter minimum), and how these questions might be addressed using tracking algorithms.

3. Choosing 500 hPa as the main level in which the analysis is done is reasonable, but the authors should add a figure describing the performance of the algorithm as a function of tracking level to give a more general representation.

4. The effect of ENSO and the midwinter suppression on the number of eddies is crucial in understanding their overall effect. Although it is challenging to measure as it appears differently depending on the tracking algorithm and the parameter that is tracked. Therefore, the effect of ENSO and the midwinter suppression on the number of tracked eddies as found by the described algorithm should be shown (either in figures 5 and 6 or as additional figures).

5. One of the most commonly used Lagrangian methods is creating composites based on the tracking data. Using this algorithm in order to make composites might be even more useful as the algorithm finds the region associated with the systems, and improved composites can be made. Adding a composite analysis of some field might be

very interesting (for example, a composite of the vertical velocity can be made and later compared to the more complicated analysis using LAGRANTO).

6. The analysis shown in figure 7 is confusing. These eddies are, to a good approximation, a closed system and therefore mass (and PV) conserving. Looking at the total vertical movement of parcels does not necessarily mean much and the conclusion that vertical processes play a small role in the dynamics might be incorrect. Previous papers (e.g., Booth et al., 2015; Tamarin-Brodsky and Hadas, 2019) have shown (as in figure 8), that there is a small section of rapidly ascending air and a larger section of moderately descending air.

Minor comments:

1. Figure 1 can be reduced to 3 subfigures, combining subfigure b-c and e-f and describe the system's age in the text.

2. The use of the phrase "low- (high-) pressure system" (e.g., page 4, line 24) is confusing in this context, as it is mostly used to refer to low-level cyclones (anticyclones).

3. The climatology calculations that led to the colours and contours in figure 3 should be better explained.

4. Page 11, line 32 to page 12, line 2: These results have been discussed before in the context of extratropical storms, both in simulations (Booth et al., 2015), and observations (Tamarin-Brodsky and Hadas, 2019). A reference should be made.

Typos

1. Page 3, line 17: "The trough and ridge identification algorithm is based on...". Perhaps use "The trough and ridge identification are done on..."

2. Page 4, line 20: Number is unclear.

---

## Author Comment (AC1) · 5 Jun 2020

"The Life Cycle of Upper-Level Troughs and Ridges: A Novel Detection Method, Climatologies and LagrangianCharacteristics"

**Reply to review #1**

**We appreciate the positive comments from the Reviewer and also the critical and detailed comments made, which have led to an improved manuscript. In response to the comments made by you and the other Reviewers, we have made considerable changes to the text and figures, and we will mention the relevant ones in our point-by-point response. We also uploaded a tracked-changes version of the manuscript.**

**Reviewer:** My biggest concern is with the authors' choice of 500 hPa geopotential height (Z500) as the variable for analysis of so called "upper-level" troughs and ridges. As far as I can see, no justification is provided for the choice of this variable. Historically Z500 was widely used in the 1970s and 1980s for analyses of troughs and ridges. However, since PV thinking became mature (cumulating in the seminar paper by Hoskins et al. 1985), synoptic dynamicists have generally accepted that troughs and ridges are manifestation of PV anomalies that have largest amplitudes either at the tropopause or at the surface, and in recent decades, most analyses have focused on analyzing variables either near tropopause level or near the surface (e.g. Hoskins and Hodges 2002, Fig. 1). In this paper, the authors pick the mid-level (500 hPa) for analyses of upper level troughs. Can the authors please provide justification why they pick a mid-troposphere variable instead of an upper troposphere varable to analyze?

**Authors:** The 500 hPa geopotential height (Z500) has traditionally been used for analysis of troughs and ridges. It is still used amongst forecasters and researchers when describing the synoptic-scale weather evolution in terms of high- and low-pressure systems, and troughs and ridges. Because of its historical significance in synoptic meteorology, we have chosen Z500 in this study for demonstration purposes. Depending on the specific research question, researchers may choose other suitable variables or levels as input data. For example, potential vorticity (PV) or potential temperature on the dynamical tropopause. The use of Z500 in our manuscript is not a statement on its superiority over other variables. The equation used for the computation of the quasi-geostrophic omega forcing however, which is shown in several figures throughout the manuscript, relies on the geopotential height. The combination of the quasi-geostrophic forcing of vertical motion and the geopotential-based troughs and ridges yields a powerful research tool in addition to its use in descriptive synoptic meteorology.

- We supplement the revised manuscript with the results for Z300 (as supplementary figures) and comment briefly on changes between Z500 and Z300 where appropriate.
- We further added a statement that clarifies the use of Z500 for demonstration purposed and that the user of our tool is free to choose other suitable variable or levels depending on the exact research question.

**Reviewer:** Also, why use curvature of geopotential height contours? Why not use relative vorticity or PV instead? Given that dynamically, we can easily write equations for vorticity or PV tendency while it is difficult to write an equation governing the tendency of curvature of Z500 contours, what are the advantages for picking such a variable to analyze? Can't trough/ridge axes be defined based on vorticity or PV?

**Authors:** Our approach is to identify troughs and ridges using geometric means in an effort to mimic the human eye. Geometric means have successfully been used to identify streamers (e.g., Wernli and Schwierz 2006), Rossby wave breaking (e.g., Barnes and Hartmann 2012, and many others), Rossby wave initiation (Röthlisberger et al. 2015) and also surface cyclones and anticyclones (closed SLP contours are used in many routines; Neu et al. 2013). Potential vorticity and its curvature could also be used as input data and the user of our tool is free to choose PV instead of Z. In general, vorticity could also be used, but given its rather uneven and noisy distribution over neighboring regions it comes with its own difficulties. In high-resolution NWP data, in particular, the many small-scale relative vorticity maxima become a hindrance in unambiguously identifying troughs and ridges. This is somewhat similar to the challenges of identifying cyclones in high-resolution data based on relative vorticity.

**Reviewer:** A potential issue is that results can be affected by high terrain, e.g. the Tibetan Plateau. Previous studies (e.g. Chang and Yu 1999, Hoskins and Hodges 2002 (HH02 hereafter); Hakim 2003) have shown that there are upper level waves that propagate along the subtropical jet in winter near the southern edge of the Tibetan Plateau. These waves are clearly missing from this study (Fig. 3a and 4a). These waves are potentially important in understanding Pacific cyclongenesis (e.g. Chang 2005) and the mid-winter suppression (Nakamura and Sampe 2002).

**Authors:**

(i)    The 500-hPa indeed intersects the Tibetan Plateau and other high mountain ranges such as the Rocky Mountains and the Andes. Caution must therefore be given when interpreting the results over these regions. Troughs and ridges are therefore not computed at times and places where missing data values are present in the data.

(ii)    It is correct that the North Pacific storm track is fed by two upper-level seeding branches, one coming from the Siberian storm track and another one along the subtropical jet (HH02, Chang 2005). However, it is the northern seeding branch that dominates surface cyclogenesis between Nov–Mar (Hakim 2003, Chang 2005 see discussion on p. 2004/05). The waves along the southern branch are best seen in bandpass-filtered data at 300-hPa along 25° S, typically in composites preceding cases of rather deep cyclogenesis (Chang 2005). Sometimes both seeding branches even interact (Fig. 4 in Chang 2005). In the seasonal mean climatology, we do not see these waves at the 500-hPa level. Most likely the key difference is the time filtering, which excludes waves that act on time scales longer than the typically synoptic times. Thereby these climatologies emphasize a subcategory of

eddies, which otherwise are masked in unfiltered climatologies. This subcategory of waves however is still part of our data, as is shown below for a trough-ridge train along the southern seeding branch between 60–90° E and near 120° E on 26th January 2017. The downstream trough and ridge even connect the northern with the southern seeding branch, which is in agreement with Fig.4 in Chang (2005). In the revised manuscript, we placed a comment on this in the corresponding section.

[Figure]

*Figure 1: Troughs (blue) and ridges (red) along the southern seeding branch on 26th January 2017.*

**Reviewer:** While the authors discussed some consistencies with previous studies for summer climatology (section 3.2), for the much more researched winter, they didn't provide much comparisons with previous studies. More reference to previous studies should be made in section 3.1. There are some differences that could potentially be due to differences in methodologies. For example, comparing their Fig. 3a to Fig. 11d of HH02, it is apparent that the authors' results are quite a bit further north than those presented in HH02. Also, the authors' results (Fig. 4a) show nearly absence of ridges to the south of Japan, while HH02 shows that 250 hPa Z positive tracks have a maximum south of Japan over western Pacific. Can the authors please explain what might be the reason(s) behind such differences? In any case, comparisons with results from previous studies should be discussed more.

**Authors:** Thank you for pointing us towards this imbalance between the discussions of the summer and winter climatologies. In the revised version, we now make more comparisons with previous studies, in particular with climatologies of cyclone frequencies and streamers. We do not expect Z500-based troughs and ridges to yield similar climatologies as for bandpass filtered 250-hPa Z eddies (HH02), see our discussion of the southern seeding branch over the Pacific above.

The more poleward location of our climatology (Z500) compared to Fig.11d in HH02, which is based on bandpass-filtered 850-hPa meridional wind, relates to the general westward and poleward tilt with height of baroclinic waves and the relative location of the 500-hPa trough to the surface warm sector.

Note that there is a good agreement between the genesis and lysis hotspots found by HH02 (Fig.5c, d) with the trough and ridge frequencies in our climatologies.

**Reviewer:** In my opinion, the methodology is not described in sufficient details. How the tracking was done was not really presented and readers are referred to a Ph.D. dissertation (221 pages long). This seems to me an important step since one of the parameters the authors emphasized is age of the system which depends critically on the tracking algorithm. Details of how tracking is done should be presented - perhaps as an appendix or supplemetal material.

**Authors:** In the revised version, we added a full new and detailed paragraph on the feature tracking in the method section. We also point the reader towards specific pages in the Ph.D. thesis were necessary.

**Reviewer:** As the authors mentioned, there are always rather arbitrary cutoff values in any objective algorithms. For this algorithm, they mentioned three - the minimum curvature, the size of the trough/ridge objects, and the length of the axis. How sensitive are the results of these choices? How are these choices set? For example, in the example shown in Fig. 2, the trough over the Mediterranean in panel 2a looks to me a very legitimate trough, but it was not identified. I would guess that even 6 hours earlier that trough was already apparent. So, the question is: what is the justification for the authors to pick those particular cutoff values? This is important for genesis and lysis, and for trough ages and lifetimes. In my opinion, sensitivity to these cutoff parameters should be explored and discussed.

**Authors:** There is unlikely an ultimate answer to this question. Even trained experts will not agree in every situation on the presence of a specific flow feature, which is a lesson learned from the discussion surrounding the detection of fronts (e.g., Schemm et al. 2018), but we do agree that it is important that the cutoff values are discussed. In general, the patterns seen in the winter and summer climatologies (Figs. 3 and 4) are robust. Reducing/increasing the thresholds will emphasize or de-emphasize the centers of action, i.e., detection frequencies will increase or decrease, but they will not change locations. Note also that the thresholds might easily be adjusted for individual case studies to obtain an optimal match to a subjective analysis. This, however, is far more challenging if a climatology of troughs and ridges is compiled. On the other hand, the amplitudes in a climatology might be sensitive to the chosen thresholds to some degree, but we expect the general pattern to be rather robust.

The example in Fig. 2 was *not* used for the tuning of the cutoff values in the algorithm. It was chosen intentionally because we want to present with maximum transparency the difficulties arising from the need to define these cutoff values. This transparency would be missing if we would present a "perfect" case that was used for the tuning (in fact we used, among others, the cases shown in Bue and Xie (2015)). If we reduced the cutoff values to catch the trough already 6 hours earlier, we could almost certainly identify another situation where we would tend to increase the cutoff value again. We believe that it is important that the cutoff values are openly communicated and they are discussed in the method section and also in the illustrative case example.

- Bueh, C. and Z. Xie, 2015: An Objective Technique for Detecting Large-Scale Tilted Ridges and Troughs and Its Application to an East Asian Cold Event. Mon. Wea. Rev., 143, 4765–4783, https://doi.org/10.1175/MWR-D-14-00238.1
- Schemm, S., M. Sprenger, and H. Wernli, 2018: When during Their Life Cycle Are Extratropical Cyclones Attended by Fronts?. Bull. Amer. Meteor. Soc., 99, 149–165, https://doi.org/10.1175/BAMS-D-16-0261.1

**Reviewer:** Many objective tracking algorithms (e.g. HH02) employ other cutoff values, like feature lifetime and distance travelled. As far as I can see no such cutoffs are employed here. Could the statistics be heavily contaminated by very transient short-lived features like heat lows, or quasi-stationary climatological features?

**Authors:** We believe that it is natural for quasi-stationary features to dominate the climatology and we regard their presence not as a contamination. They are legitimate members of the trough and ridge family. The transient troughs and ridges are also contained in our data. If necessary, to address a specific research question, the stationary features can easily be removed from the data in a post-processing step or bandpass filtered input data could be used.

Heat lows are excluded. These features are associated with closed contours at the 500-hPa level and are removed from the data. Additional cutoff values like lifetime and distance travelled, which are invoked by some cyclone tracking schemes, are legitimate and can be useful, but they will introduce new degrees of freedom resulting in the same need for a justification as discussed above (How far travelled? How long-lived?). In fact, to obtain the frequencies for troughs with an age between 0-24 hours (Figs. 3 and 4), the trough objects are filtered according to their lifetime.

- The revised manuscript contains a supplement with the trough and ridge climatologies but with short-lived objects removed (a minimum lifetime of 24 hours is required) – see Fig. S3

**Reviewer:** Some of the statistics shown are not clearly defined. For example, what does trough detection frequency mean? Is that defined based on trough axis or trough objects? Also, what does the "selected frequencies of troughs with an age between 0 and 24 hours (yellow)" in the figure caption of Fig. 3 mean? Without clear definitions of these parameters it is difficult for readers to interpret these figures. Mathematical definition should be provided.

**Authors:** Thank you for pointing this out.

(i)    Detection frequencies are based on the 2d objects. Grid points outside every coherent 2d object are flagged with zeros, grid points insides with ones. Time-averaging over the obtained binary fields yields detection frequencies, which indicate the fraction of time steps affected by a trough or ridge relative to all time steps. We now provide this information in the method section.

(ii)   (ii) The single yellow contour denotes the frequency that is half the maximum detection frequency for troughs with an age between 0 and 24 hours. To this end, we

remove all 2d trough and ridge objects that are older than one day and re-compute the detection frequency based on the remaining troughs. The yellow contour is half the value of the maximum detection frequency of the thereby filtered troughs.

**Reviewer:** For the trough age, it is surprising to see that eastern Pacific/Gulf of Alaska is apparently a region where there is high frequency of 0-24-hour troughs (Fig. 3a). I would imagine only regions where there is frequent trough genesis would be highlighted, and that is certainly not a region that is associated with frequent trough genesis. As far as I can see the authors did not mention that region. Can the authors please discuss that?

**Authors:** The Gulf of Alaska is in agreement with climatologies of streamers (Fig.3 in Wernli and Sprenger 2007) and cycloysis (Fig.5d in Hoskins and Hodges 2002). These references are now mentioned in Section 3.1. Our interpretation is that trough genesis in the Bay of Alaska is resulting from wave breaking and consecutive downstream development during the decaying phase of mature extratropical cyclones over the central Pacific. In this sense, the Bay of Alaska shares some similarities to the trough genesis region over the UK, which, in our opinion, is also a follow-up development driven by upstream decaying mature extratropical cyclones over the eastern North Atlantic.

**Reviewer:** Apart from the Gulf of Alaska, it seems that "young" troughs and ridges are most frequent over regions with high trough and ridge frequencies (Fig. 3 and 4). Those regions are expected to be regions where troughs and ridges are quasi-stationary and hence presumably "old" rather than "young". Nevertheless, it depends on what the yellow contour really shows as it is not really defined, but this point should be further discussed.

**Authors:** This is a good point. Downstream of mountains troughs are frequent and thus quasi-stationary, but downstream of mountains troughs are also generated due the flow across the mountain ridge. We believe it is not too surprising that downstream of mountains a trough genesis region is identified. Locally, both the genesis and lysis frequencies are large and troughs growth and decay depend on the flow across the mountains, but they remain stationary at the same time.

**Reviewer:** In the Lagrangian analysis, one of the authors' goals is to quantify the diabatic impact (p. 14, line 8). My question is whether 500 hPa level is the best level to do that? It is still within the cloud layer at a location where strong heating is still going on. Wouldn't the full impact of diabatic heating be clearer at the tropopause level?

**Authors:** A comparison between the results based on 500-hPa and 300-hPa troughs and ridges would indeed be insightful. We agree that the full diabatic impact integrated across the entire depth of the troposphere will be more complete if the parcel trajectories were released at a higher level. However, we expect to catch a major part of the condensation, which peaks typically between 850–700 hPa, and also the below-cloud evaporation. Indeed, the trajectories will to a much lesser extent experience diabatic modification due to, for example, freezing. However, if consideration is given to diabatic modification of 500-hPa trough and ridges, it seems natural to release the parcel trajectories from this level.

Overall, this section gives a unique look into a potentially powerful combination of two tools (troughs & ridges plus parcel trajectories), which opens new possibilities for exciting research, but this proof-of-concept is far from being compete. An in-depth analysis would also require several years of trough, ridge and trajectory data and a comparison between trajectories of different lengths and released from different levels or at different trough ages. This would be a study of high scientific merit, in particular if diabatic tendencies from different microphysical processes were available, but a full-fledged stand-alone study on its own would be needed.

**Other Concerns:**

**Reviewer:** i) p. 4, line 21: 20S-70N: The authors discussed that they focus on mid-latitude troughs/ridges, but apparently did the analysis in the tropics also. In the tropics, wouldn't the fields be very noisy given the weak geopotential gradient? Why is it necessary to perform the analysis down to 20S?

**Authors:** The frequency of trough detection drops to zero between 20–25S, the tropics are therefore climatologically not relevant. However, sometimes streamers might elongate far equatorward and we think it is useful to catch those cases as well.

**Reviewer:** ii) p. 6, lines 4-5: I'm not sure I understand what the authors meant by the sentence "Interestingly here it is a trough over the North Atlantic and the ridge downstream al- ready exists for a longer time period than the up- and downstream troughs". Which troughs and ridges are they referring to in these descriptions?

**Authors:** We removed the first part of the sentence, it now reads "Interestingly, the ridge downstream already exists for a longer time period than the up- and downstream trough". The synoptic situation appears to be a downstream-development scenario; however, we would expect the upstream trough to be the oldest feature followed by the downstream ridge followed by the next downstream trough. The ridge however is the oldest features, which we did not expect.

**Reviewer:** iii) While the trough/ridge tilt is indeed closely related to the E-vectors, fundamentally tilted troughs/ridges are related to eddy momentum fluxes (which make up the E-vectors) and have been discussed as early as Jeffries (1926) and Starr (1948).

**Authors:** The fact that the tilt orientation and the corresponding orientation of the E-vector are fundamentally a result of the eddy momentum flux is mentioned several times throughout the manuscript. In the revised version, we now also define the E vector, and point the reader to the two original papers by Jeffries (1926) and Starr (1948). Thank you for mentioning these two seminal studies.

**Reviewer:** iv) p.8 line 25: Here the authors write that there is a second maximum near Lake Baikal, which to me is not really accurate since Lake Baikal apparently is located between two relative maxima, one west of 90E and he other over eastern Siberia, northeastern China.

**Authors:** Yes, corrected and changed to "west of 90E".

**Reviewer:** v) Fig. 5: Are the anomalies shown statistically significant? Some of the anomalies seem rather small.

**Authors:** The anomalies are in good agreement with the anomalies seen in E vectors (Drouard et al. 2015) and we therefore consider them physically meaningful and a good indicator for a consistent change in the orientation during ENSO. The idea here is to compare the results of the two diagnostics against each other and not to show that ENSO produces statistically significant eddy anomaly fluxes. We now set regions where the trough and ridge frequency are below 2% to missing data value.

**Reviewer:** vi) p. 10, lines 22-23, "more systems will nnow grow on the poleward flank of the jet" this should be quantified or reference cited. Both the jet and storm track shifts equatorward during mid-winter so it is not entirely clear that more systems grow on the poleward flank of the jet. Perhaps a histogram showing the frequency as a functio of tilt would quantify this.

**and** vii) p. 10, lines 25-26: Same comment as above.

**Authors:** Please find below a figure for the meridional wind shear of the zonal wind (dU/dy) over the Pacific. During midwinter (black contour), wider parts of the main growth and propagation sector of extratropical cyclones (30-60N) are affected by stronger cyclonic shear, which led us to the interpretation that all poleward tracking storms develop more consistently on the poleward flank of the subtropical jet, assuming that genesis is at a relatively fixed location over the Kuroshio. In agreement with the enhanced cyclonic shear over the Pacific (see below) are the more cyclonically oriented troughs and ridges during midwinter.

[Figure]

*Figure 2: Meridional wind shear of the zonal wind over the Pacific in winter.*

**Reviewer:** viii) p. 10, lines 31-32: "there is no marked reduction in the number of troughs and ridges" This is surprising given the strong decrease in number of troughs found in

Penny et al. Can the authors show the results and provide some explanation on why there is such disagreement between their results and those of previous results?

**Authors:** This result is an agreement with Schemm and Schneider (2018) who note that the number of surface cyclones in the central Pacific is not decreasing during midwinter. Penny et al. (2010) analyzed upper-level eddy activity at the 300-hPa level in spatially and temporally filtered geopotential height. As is the case with many measures, the midwinter suppression is seen in bandpass-filtered data (eddies and related eddy quantities) but not in measures of the mean flow (mean baroclinicity, mean zonal wind at 300 hPa). It seems as if features identified in raw data, without any frequency filtering (surface cyclones in SLP, Z500 troughs and ridges), exhibit no well-marked suppression, while frequency-filtered measures do. We cannot guarantee that this is the answer, but it appears to be the main methodological difference. In the revised version we added the detection frequencies into the corresponding panels as yellow contours. We find a new localized detection maximum over the eastern North Pacific, which potentially relates to wave breaking and lysis, since detection maximum show a good agreement with cyclolysis frequencies (Fig.5 in HH02). This underpins the presented hypothesis of accelerated, or interrupted, life cycles during midwinter over the eastern North Pacific.

**Reviewer:** ix) p. 11, line 6: "a fast and intense deepening phase followed by a rapid decay". The familiar LC2 lifecycle of Thorncroft et al (1993) shows a cyclonic lifecycle that decays very slowly. Perhaps the authors should provide references that show rapid decay of cyclonic lifecycles.

**Authors:** Yes, it is in fact the LC1 that shows an EKE peak followed by a rapid decay (Fig. 4 in Thorncroft et al. 1993). We find however more cyclonically oriented troughs but the reduction in cyclone lifetime was shown in Schemm and Schneider (2018). We thus agree that the familiar and idealized LC2 concept seems not to apply. The LC2 life cycle eventually gets dismantled.

**Reviewer:** x) p. 12, line 14: Should refer to Fig. 9 here.
**Authors:** Included.

**Reviewer:** x) p. 12, line 14: Should refer to Fig. 9 here.
**Authors**: Corrected.

**Reviewer:** xi) p. 12, lines 19-20: The windspeed in summer is also weaker so even with the same tilt the vertical motion would still be reduced. This again is related to weaker baroclinicity through the thermal wind relation but still this should be mentioned. Both reduced tilt and reduced wind speed can contribute to decrease in vertical displacement.

**Authors:** Good point, we added this argument.

.

---

## Author Comment (AC3) · 5 Jun 2020

"The Life Cycle of Upper-Level Troughs and Ridges: A Novel Detection Method, Climatologies and Lagrangian Characteristics"

**Reply to review #3**

**We would like to thank Reviewer #3 for the specific and positive comments on how we can improve the presented manuscript. In response to the comments made by you and the other Reviewers, we have made considerable changes to the text and figures, and we will mention the relevant ones in our point-by-point response.**

**Reviewer:** As one of the most important skills of this tracking algorithm is finding the axis of the system, and a lot of the analysis is based on it, more effort should be invested in explaining the notation of cyclonic and anticyclonic tilt, what are the factors controlling the axis orientation and the effect of the axis orientation on the life cycle of the system and it's interaction with lower-level systems. A cartoon demonstrating these concepts might be helpful.

**Authors:** The ideal schematic to present the notion of cyclonic and anticyclonic tilt is in our opinion Fig. 12 of Thoncroft et al. (1993), which is shown below. We now reference explicitly this schematic in our introduction.

Further, we added a new Figure that explains the algorithm and, in that schematic, we added a trough axis and point the reader to the orientation of this axis to illustrate the notion of cyclonic and anticyclonic orientation (New Figure 2).

34                C. D. THORNCROFT *et al.*

Figure 12. Schematic of a PV-theta contour in an Atlantic storm track sharing its main characteristics with (a) an LC1-type life cycle and (b) an LC2-type life cycle. The dashed line marks the approximate position of the mean jet at each stage.

**Reviewer:** In the introduction, more attention should be given to explain the open questions addressed in this work (ENSO, midwinter minimum), and how these questions might be addressed using tracking algorithms.

**Authors:** Good point, we added a new paragraph to the introduction that explains the research questions related to ENSO and the midwinter suppression, which we address in our study. The main literature overview is however presented in the corresponding sections to keep the introduction concise.

**Reviewer:** Choosing 500 hPa as the main level in which the analysis is done is reasonable, but the authors should add a figure describing the performance of the algorithm as a function of tracking level to give a more general representation.

**Authors:** Ok, we added the results of the 300 hPa level as a supplement figures and now mention the possibility for the user to choose the level and variable.

**Reviewer:** The effect of ENSO and the midwinter suppression on the number of eddies is crucial in understanding their overall effect. Although it is challenging to measure as it appears differently depending on the tracking algorithm and the parameter that is tracked. Therefore, the effect of ENSO and the midwinter suppression on the number of tracked eddies as found by the described algorithm should be shown (either in figures 5 and 6 or as additional figures).

**Authors:** With respect to the influence of ENSO, we focus on the change in the orientation because this was previously inferred based on **E** vectors. We would like to demonstrate the correspondence of the two perspectives. With respect to the midwinter suppression, we fully agree, here the number/frequency of identified objects matters and we added the detection frequencies as yellow contours to the revised Fig. 6.

**Reviewer:** One of the most commonly used Lagrangian methods is creating composites based on the tracking data. Using this algorithm in order to make composites might be even more useful as the algorithm finds the region associated with the systems, and improved composites can be made. Adding a composite analysis of some field might be very interesting (for example, a composite of the vertical velocity can be made and later compared to the more complicated analysis using LAGRANTO).

**Authors:** Yes, composites are indeed a major application, which we also foresee for the troughs and ridges. Troughs and ridges could be centered on cyclogenesis or extreme events. We prefer not to add another figure to the paper because the number of Figures is already large. We now have already 10 Figures (1 new, 2 modified) plus 3 new supplementary Figures. We believe that a composite analysis deserves a full-fledged study.

**Reviewer:** The analysis shown in figure 7 is confusing. These eddies are, to a good approximation, a closed system and therefore mass (and PV) conserving. Looking at the total vertical movement of parcels does not necessarily mean much and the conclusion that vertical processes play a small role in the dynamics might be incorrect. Previous papers (e.g., Booth et al., 2015; Tamarin-Brodsky and Hadas, 2019) have shown (as in figure 8), that there is a small section of rapidly ascending air and a larger section of moderately descending air.

**Authors:** The trough in Fig.7 is classified as an open trough, because the corresponding geopotential isolines (blue contours in Fig. 7) are not closed. The closed area in Fig. 7 is the region satisfying the curvature threshold (red contours in Fig. 7). We therefore expect a continuous flow in and out of the trough area, which is suggested also by the trajectories. We are not sure in which sense the through constitutes a closed system, for a truly closed system we would expect the trajectories to circle to some degree inside the closed contour.

We however fully agree that analysis of mean quantities can be misleading in particular if the spread among the air parcels is large. Therefore, we added two new histograms to Fig. 7, which show the binned 24-hour changes in PV and pressure. PV changes are for 90% of all air parcels strongly confined to -0.2 to 0.2 PVU/24h. Thus, it is fair to argue that the flow is almost adiabatic with a small fraction of air parcels undergoing diabatic modification (of up to -0.6 PVU/24h). Pressure changes indeed reach up to 60 to 80 hPa/24h for approximately 15% of all air parcels, while ascent of up to -30 to -40 hPa/24h is found for less than 5%. It is therefore correct, as you point out, that embedded in the large-scale descending motion there is a very small fraction of ascending air. Similar examples occur for convection embedded in warm conveyor belts (Martínez-Alvarado et al., 2014; Rasp et al. 2016; Oertel et al. 2019), which itself is the more rapidly ascending fraction of air inside the rising warm sector of an extratropical cyclone. We comment on this in the revised manuscript.

**Reviewer**: Figure 1 can be reduced to 3 subfigures, combining subfigure b-c and e-f and describe the system's age in the text.

**Authors:** This is correct. We will check with the copy editor if the figure is too large and reduce it to 3 subfigures but if not, we prefer to keep it.

**Reviewer**: The use of the phrase "low- (high-) pressure system" (e.g., page 4, line 24) is con- fusing in this context, as it is mostly used to refer to low-level cyclones (anticyclones).

**Authors:** Yes, this cyclone is also well-marked at the surface. We now mention this in the text, which hopefully clarifies the wording.

**Reviewer**: The climatology calculations that led to the colours and contours in figure 3 should be better explained.

**Authors:** We tried to improve the explanation.

**Reviewer**: Page 11, line 32 to page 12, line 2: These results have been discussed before in the context of extratropical storms, both in simulations (Booth et al., 2015), and observations (Tamarin-Brodsky and Hadas, 2019). A reference should be made.

**Authors:** Yes, the asymmetry is not an entirely new finding. We added the studies of Sinclair et al. (2020), Tamarin-Brodsky and Hadas (2019), Booth et al. (2015), plus the original literature of O'Gorman (2011), who first explained the asymmetric distribution of vertical motion as a result of the influence of moisture and heating on the static stability. We would argue that the asymmetry is also resulting from moist convection being mostly upward in the atmosphere and not downward.

**Reviewer**: Page 3, line 17: "The trough and ridge identification algorithm is based on...". Per- haps use "The trough and ridge identification are done on..."

**Authors:** The corresponding sentence was changed.

**Reviewer**: Page 4, line 20: Number is unclear.

**Authors:** The number is the trough/ridge area.

---

## Author Comment (AC2)

"The Life Cycle of Upper-Level Troughs and Ridges: A Novel Detection Method, Climatologies and LagrangianCharacteristics"

**Reply to review #2**

**Dear Irina,**
**Thank you very much for your time and effort you put into this review. We very much appreciate the comments that helped us to improve the paper. Please find below our specific replies.**

**Reviewer:** My major comment has already been picked up by another reviewer: I would also like to see justification of the geopotential level used to identify troughs/ridges and, possibly, how results differ between levels. I can see a point in using an Hgt level as opposed to, e.g., PV, as Hgt levels are readily available in most model outputs, making this approach useful for analysis of, e.g., CMIP6 models. However, troughs/ridges are often seen as upper-level phenomena, while 500 hPa lies in the middle of the troposphere.

**Authors:** Geopotential height at the 500 hPa level has traditionally been used for analysis of troughs and ridges. It is still widely used for forecasting purposes and also in research. Because of its historical significance, we have used it in this study for demonstration purposes.

In the revised version, we added the results for the 300 hPa level as a supplementary figure and we also comment on notable differences where appropriate. The combination of the quasi-geostrophic forcing of vertical motion by the Q vector, which is formulated using geopotential height, and the geopotential-based troughs and ridges result in a powerful research tool. We see however also the merits of using potential vorticity for input data. The user of our research tool is free to choose any other suitable variable or level as input data depending on the specific research question to be addressed.

**Reviewer:** Abstract: It is said that tracking allows analysis of evolution of troughs/ridges. However, in my view, this paper does not present such analysis. Either remove this statement from the abstract (it can be kept in the description of the method) or add analysis of their life cycle.

**Authors:** It is correct, we do not show an explicit example in this manuscript. We changed the sentence in the abstract to "(…) *could be* used to study the temporal evolution of the trough or ridge orientation". Thanks for pointing this out.

**Reviewer**: p1, l.21-25: Add a figure showing a trough and a ridge.

**Authors:** Good idea, we now reference Fig.1.

**Reviewer**: p2, l.18: define E vector here.

**Authors:** We added the E vector definition.

**Reviewer**: p.4, l.9-16: This part needs to be illustrated - draw a figure showing vectors discussed here

**Authors:** We added such a figure.

**Reviewer**: p.5, l28: give a brief description of Q vector here first, not only a reference.

**Authors:** We added a brief description of the Q vector.

**Reviewer**: p.6, l22: can the absence of troughs be a result of northern boundary at 70N? Could it be that troughs extent further north and the part to the south of 70N is below the threshold on the length?

**Authors:** Our statement that troughs are absent at the end of the North Atlantic storm track was inaccurate. It is the center over the UK and over eastern Europe that is associated with development at the end of the North Atlantic storm track. We corrected this statement.

**Reviewer**: p.5, l. 32: The map at 06 UTC also suggests a significant trough in the Mediterranean, but it remains undetected. Is it due to the thresholds? In this case, some sensitivity test should be shown to justify the selection of thresholds.

**Authors:** The case study was chosen intentionally to highlight the difficulties arising from the need to define thresholds. The presented case was not used for the tuning of the thresholds. There is unlikely an ultimate answer to this question, because when we decrease the threshold in this case, we will find another where we would tend to increase it again. We do not want to show the perfect example in our study. We want to be transparent about this issue. We discuss for this reason the role of the thresholds in the corresponding section. In general, the patterns seen in the climatologies are robust, but the amplitude of detection frequencies depend on the used threshold. This holds true for all object-based feature detection algorithms.

**Reviewer**: p.6, l. 17: Is it frequency of trough area of trough axes?

**Authors:** Here we use the trough objects corresponding to the through area. We added this information.

**Reviewer**: p. 7, l.1: To confirm that Greenland troughs are more transient can you built pdfs of trough lifetime characteristics (propagation speed and lifetime) in regions of interest (over Greenland, Pacific that is discussed later and a reference region).

**Authors:** The Figure below shows the winter climatology of the trough lifetime at each grid point. The lifetime at Greenland is reduced compared to downstream of the Rocky Mountains or compared to East Europe and Russia. We added the figure as supplement to our manuscript.

[Figure]

Figure 1: Mean lifetime of troughs during the extended winter period (Nov–Mar).

**Reviewer**: p7, l28-30: Given the difference in frequency of troughs and ridges, did you think of applying different thresholds for their identification. Troughs and ridges can be seen as advection of cold and warm air and, due to the variable air density, typical curvature of hgt may vary.

**Authors:** This is a very interesting suggestion, but we do not find a difference in the curvature between through and ridges. Climatologically, the curvature varies between the lower threshold of 0.05 degrees/km to 0.13 degrees/km for troughs and ridges. In midlatitudes both have a mean curvature of 0.1 degrees/km. However, we did some experiments with a latitude dependency of the thresholds, but did not consider this approach further as it is mainly relevant for regions that we have excluded (<20S and >70N) from our data.

**Reviewer**: p.8: Can you think of reasons why there is no cyclone maximum near the Iberian Peninsula associated with a peak in trough frequency and Rossby wave breaking? I suggest that these conditions in the upper (middle) troposphere produce cut off lows not visible on the surface.

**Authors:** Yes, we agree. It is likely cut-off low formation that is taking place in this region. We added this information.

**Reviewer**: p.8, l34: Despite equatorward shift of troughs off the West coast of the US in summer, the ridges shift poleward during summer (something that one would have expected). Can you discuss what may cause summer poleward shift of troughs in this region?

**Authors:** We assume that PV streamer formation is more common in this region. Consider the schematic below from Thorncroft et al. (1993). If PV streamer formation, or cut off

formation, is more common (upper panel), the more equatorward elongating troughs will result in a more equatorward displaced.

Figure 12. Schematic of a PV-theta contour in an Atlantic storm track sharing its main characteristics with (a) an LC1-type life cycle and (b) an LC2-type life cycle. The dashed line marks the approximate position of the mean jet at each stage.

**Reviewer**: p.10, l33: The text refers to Fig 6, which shows mean orientation of ridges and troughs. This plot is supposed to illustrate the 'dominance' of those systems in the Pacific, however, I think the frequency should be shown instead.

**Authors:** Ok, we corrected the sentence since "dominance" is misleading. We want to highlight however the fact that the preferred orientation is now cyclonic over most parts of the Pacific, while this is not the case during the shoulder months. We do not refer to the frequency, but to the preferred orientation regardless of the frequency.

**Reviewer**: p.11: Did you test longer backward trajectories? Do troughs/ridges have life cycles? It will be nice to see some analysis on this. Are back trajectories in fig. 7 b,c sensitive to the stage of their development?

**Authors:** We did not test longer backward trajectories, but it is good to see that this section triggers a lot of interesting questions and further suggestions from both reviewers. We consider longer trajectories to be in particular useful for Z300-based troughs and ridges.

We agree that troughs and ridges have a life cycle. However, while we do know their age, it appears to us not as particularly meaningful to release the trajectories from features of similar age. Cyclones, for example, are often categorized according to their genesis, maximum deepening, maximum intensity, maximum decay and lysis. Such a definition can be based on, for example, changes in SLP along each cyclone track. We do not have a similar convention for troughs and ridges. A proper definition of the life cycle stages of troughs and ridges would be required on which the community agrees. This section "only" explores the now new possibility of using the trough and ridge objects as trajectory start location in a "proof-of-concept" fashion. It is not indented to be a complete analysis. We agree that a complete analysis would be very interesting, with different levels and different trajectory lengths, for a complete 30-year period and for different life cycle stages on both hemispheres. All of these questions would easily fill one or two completely new studies and we cannot perform them in this paper.

**Reviewer**: p.12: A histogram of north/south displacements might be interesting. It is worse noticing that fig. 8a,b and e,f are in good agreement with each other, i.e. decent

corresponds to a reduction in potential temperature. However, then the statement that the downward motions occur along isentropic surfaces sloping equatorward is questionable (though I like to think that the flow is along isentropes).

**Authors:** We agree, there is on average a mild decrease in potential temperature so the motion is not fully isentropic, but quasi-isentropic. This reduction is found along trajectories that exhibit also a weak downward motion of 0-100hPa in 24 hours starting from the 500 hPa level and we therefore speculate that this is due to radiative cooling. We corrected this statement in the revised manuscript.

**Reviewer**: p.12, l2: what is the difference between a trough with closed geopotential isolines and a cyclone at that level?

**Authors:** Yes, it could be a deep surface cyclone or cut-off low with no surface signature, but we want to treat those distinct from "pure" troughs and ridges and the algorithm thus labels the former as closed-contour or closed troughs and the latter and open troughs. Although the distinction between closed and non-closed (open) troughs is somewhat arbitrary, it also matches with a human forecaster's perspective. And, as stated in the manuscript, it is also partly the aim of our approach to mimic the geometric view of a human forecaster.

**Reviewer**: p14, l.10: Is the code available online?

**Authors:** We are happy to share the data and the code. As is the case for all our feature-based climatologies, the data will be provided at [http://eraiclim.ethz.ch](http://eraiclim.ethz.ch) (see Sprenger et al. 2017 for a full list of available climatologies; note that we are in the transition to ERA5 data) and the Fortran and Python codes will be made available on request.

- Sprenger, M., et al. 2017: Global Climatologies of Eulerian and Lagrangian Flow Features based on ERA-Interim. Bull. Amer. Meteor. Soc., 98, 1739–1748, [https://doi.org/10.1175/BAMS-D-15-00299.1](https://doi.org/10.1175/BAMS-D-15-00299.1)

**Reviewer**: Abstract: "methods that detect the initiation phase of upper-level Rossby wave development" - did you mean Rothlisberger et al. 2016 paper or another? I could not find a paper on this in the references.

**Authors:** Yes, this was missing, we added a reference to the introduction.

**Reviewer**: p1, l.22: for a good reason.

**Authors:** Corrected.

**Reviewer**: p1, l.24: replace enchanted with increased.

**Authors:** Corrected.

**Reviewer**: p2, l.3: re-phrase to read better.

**Authors:** We tried to simplify the sentence.

**Reviewer**: P6, l19: change to 'Altai' (multiple times in the paper). Not sure whether this is still east Asia. Perhaps, Far East is better.

**Authors:** We changed the wording consistently to "Altai". Maybe Far East is a bit ancient. We removed East Asia as suggest and no consistently refer to Altai (the latter is partly in East and Central Asia).

**Reviewer**: P.6, l20: remove 'slightly'

**Authors:** Removed.

**Reviewer**: Fig. 3, 4: On my screen continents are barely visible. As for elevated areas, I suggest either marking them with a different colour or hatching.

**Authors:** We did some experiments with hatching, but turned out to look rather patchy in the printed version. We will make sure, together with the editing team, that during the typesetting the quality of the continents will be increased. We did not use the high-quality figures for the initial submission and will submit higher-quality figures to the editorial office.

**Reviewer**: p.11, last line: I am not sure why you say 'pushing air', I guess it is being pushed. Perhaps 'moving air' is better.

**Authors:** Yes, perhaps "moving air" is better, we replaced "pushing air" with "moving air".

**Reviewer**: p.12, l6: remove 'most'

**Authors:** Corrected.

**Reviewer**: p.12, l3: change 'ore' to 'or'

**Authors:** Corrected.

Thank you for the helpful comments!

---

## Author Response (AR2)

"The Life Cycle of Upper-Level Troughs and Ridges: A Novel Detection Method, Climatologies and LagrangianCharacteristics"

**Reply to review #1 (Report #3)**

**Reviewer:** I think the authors have responded reasonably to most of my comments, except perhaps one point which I don't quite agree with their response. Other than that, I only have a couple of minor comments.

**Major comment:**

**Reviewer:**

1) In my previous comments, I commented on the missing troughs and ridges along the subtropical jet south of the Tibetan Plateau, which are clearly present in previous studies (e.g. Fig. 14 of Hoskins and Hodges 2002, Chang and Yu 1999, and Chang 2005). In fact, if one compares the schematic in Hoskins and Hodges Fig. 14, all upper level principal tracks shown there can be found here, with the only exception being the southern branch over Eurasia. In this revision, the authors also showed results using 300 hPa height in the supplement, but those troughs/ridges are still absent at 300 hPa. In their response, the authors claimed that the waves in the southern branch are best seen in band pass filtered data, but in fact that's not the case. Chang and Yu (1999), and Chang (2005) both used unfiltered data, not band-pass filtered data as claimed by the authors in their response. Chang and Yu (1999) showed that these waves generally have longer periods (8 days or longer) than waves in the oceanic basins, and the composites shown in Chang (2005) also showed that these waves have low phase speed and longer periods. Hoskins and Hodges (2002) did use spatially filtered data (but not band-pass filtered data), but my guess is that since Hoskins and Hodges used vorticity at 250 hPa, even vorticity data that is not spatially filtered would show these systems. So, the absence of these systems, even at 300 hPa, in the authors' results is still a mystery. In their response, the authors also stated that "This subcategory of waves however is still part of our data" and showed an example of that. The question then is why they don't show up in the climatology, while they showed up clearly in climatologies of previous studies (e.g. Fig. 12f and 13b of Hoskins and Hodges 2002)? The authors stated in their response that "In the revised manuscript, we placed a comment on this in the corresponding section". I might have missed that but I don't think I saw that comment in the revised manuscript. Can the authors point out where that comment is? In my opinion, as this seems to be the main difference between results of this study to those of previous studies for winter systems, this should definitely be pointed out and discussed in the paper.

**Authors:** We would like to thank the reviewer for pointing out this issue. We agree that this needs a better clarification. Our statement that previous studies used band pass filtered data was indeed not accurate enough. Nevertheless, most previous studies remove in one way or another a background state from the input data. Chang and Yu (1999) state that they remove the stationary component of the flow and the seasonal mean climatology. Hoskins and Hodges (2002) use spatially filtered data. We assume that this could cause the difference related to the southern branch over Eurasia. However, after giving this issue again considerable thought and given the fact that these waves are sometimes detected by our

algorithm (see the example provided in our last response), the issue might also result from the cut off value of the curvature. Climatologically, the curvature of the geopotential isolines varies with latitude. The mean curvature at 25ºN over Eurasia along the southern branch is 0.07 º/km, while it is 0.1 º/km near 55ºN. The cut off value used in this study is 0.05 º/km. So, while troughs and ridges are detected at both latitudes, the reduced detection frequency along the southern seeding branch might result from the latitudinal dependency of the mean curvature of the geopotential isolines.

- In response to the reviewer's comment, we highlight the weaker detection rates compared with Chang and Yu (1999) and Chang (2005) along the southern branch over Eurasia more prominently in the main text (p. 8, lines 5–9).
- We also introduced a subsection on the limitations of our approach in the final section, where we discuss the issue with the latitudinal dependency. In an upcoming release of the algorithm, we will consider adding the possibility to use a minimum curvature that varies with latitude rather than using a fixed global threshold (p. 16, lines 25–35).

**Other comments:**

**Reviewer:**

2) p. 10, lines 16-17: Could that be related to the authors' choice of not including areas within closed contours as part of troughs?

**Authors:** The agreement between throughs and streamers is to some extent reinforced by not including areas within closed contours.

**Reviewer:**

3) p. 12, line 24: "bandpass-filtered eddies". This is also related to the authors' response to one of my previous comments. If I am not mistaken, Penny et al (2010) did not use bandpass-filtered data. They did remove the large spatial scale and long-time scale (90-day high pass), but I don't think characterizing their data as bandpass-filtered is accurate.

**Authors:** Yes, Penny et al. (2010) use a 90-day high pass filter and a spatial filter that admits only planetary wavenumbers between 5 and 42 on top of that. We corrected our statement.

"The Life Cycle of Upper-Level Troughs and Ridges: A Novel Detection Method, Climatologies and Lagrangian Characteristics"

**Reply to review #2 (Report #2)**

**Reviewer:** The authors have done an excellent job reviewing the manuscript. They addressed all major questions raised by reviewers. The new manuscript reads better and new figures help understand the method description. I find the new section on the relationship to the ENSO to be very valuable. I am happy to recommend the revised version of the manuscript for publication subject to very minor revision.

**Minor:**

1.  p.1, l.19: change to 'oriented'. Corrected.

2.  p.1, l.24: for a good reason; equatorward moving cold air. Corrected.

3.  p2, l.15: 'Under anticyclonic shear, the trough deforms into a narrow band, called a streamer, which extends equatorward to the mean jet position.' It should probably say 'which crosses the jet and extends equatorward of the mean jet position.' This type of wave breaking often results into formation of cut-off lows which may be mentioned here. Corrected and we also added the cut-off low formation following streamer formation.

4.  p.4, l.29: suggest identifying. Corrected.

5.  Fig.2 (p.24, caption): (4) a 5 km step along the geopotential height isolines in the direction of vector (3) vector and a new tangent vector (green) derived at the new position. Corrected, thank you for this suggestion.

6.  p.6, l.11: two consecutive time steps, ti and tj, are found. Added.

7.  p.6, l.14: a splitting if tj > ti and a merging if ti > tj. Corrected.

8.  P.7, l.30: Along with the low frequency of troughs next to Greenland, there is a relatively low number of ridges downstream of Rocky Mountains. Schemm et al (2018) suggests a similar number of troughs and ridges in both areas. In Schemm et al. (2018) composites of cyclogenesis are shown but not a climatology of the numbers. We added the statement on the relatively low number of Rocky Mountain ridges to the ridge section (p. 9, line 22).

[revised manuscript text omitted]

9. p.9, l. 8: remains at a relatively high level. Corrected.

10. P10, l.12: for such synoptic situation. Corrected.

11. P.10, l.25: a particular progress. Ok.

12. P.11, l.13: Can you better define what you mean by ENSO-affected winters? Is there, e.g., any threshold on the index value or a time lag? As stated in the text we simply follow the ONI definition: https://origin.cpc.ncep.noaa.gov/products/analysis_monitoring/ensostuff/ONI_v5.php

13. P.12, l.11: "The vertical eddy structure 10 – which is more poleward when the eddy efficiency decreases during midwinter" - what do you mean by more poleward structure of eddies? The orientation of the vertical tilt of the eddy geopotential isolines is not only westward with height but also more poleward. We changed the statement.

14. P.12, l.15: "The equatorward shift of the Pacific jet changes the large-scale environment in which synoptic systems grow because more systems will grow on the poleward flank of the jet." How do you know that it is not the other way round, i.e. more frequent cyclonic wave breaking shifts the jet equatorward? Here we refer not to the eddy-driven jet but to the thermally-driven subtropical jet over the Pacific.

15. Fig. S.3: '500-hPa'. Corrected.

.